# Efficient Fine-Tuning of Large Language Models with Zeroth-Order Model Parallelism

## Abstract

Model parallelism (MP) is a widely adopted paradigm for scaling large language model (LLM) training across multiple nodes. Yet, existing methods mainly rely on first-order optimization, which suffer from two key bottlenecks: *high communication overhead* due to frequent transmission of activations and gradients, and *substantial memory consumption* caused by caching these intermediate states. Zeroth-order (ZO) optimization offers a compelling alternative by eliminating explicit gradient computation and storage, naturally reducing communication and memory costs. Despite these advantages, ZO methods remain largely unexplored in the context of MP for LLM fine-tuning. In this work, we first investigate activation sparsity patterns induced by common activation functions (e.g., ReLU, GELU, SwiGLU) during LLM fine-tuning. Motivated by these key observations, we propose `SparQ`, a ZO-based MP framework that exploits quantization-induced activation sparsity to reduce memory footprint and communication overhead. `SparQ` consists of three key components: (1) using the gradient-free nature of ZO optimization to eliminate gradients; (2) applying activation quantization to induce sparsity that enables efficient sparse encoding; and (3) strategically placing split layers at sparsity-rich regions and transmitting activations in sparse form, significantly reducing communication cost with minimal impact on model performance. We theoretically establish that `SparQ` achieves a sublinear convergence rate for non-convex objectives. Extensive experiments show that `SparQ` reduces GPU memory usage by over $3\times$ and communication cost by $50\%+$ while maintaining comparable LLM fine-tuning performance across multiple tasks, compared to the MP baseline AQ-SGD.

## 1 Introduction

Large language models (LLMs) have demonstrated strong generalization capabilities, driving their adoption across diverse downstream tasks (Vaswani, 2017; Zhao et al., 2023; Naveed et al., 2025), such as privacy leakage detection (Zhu et al., 2024; Chen et al., 2026), mathematical reasoning (Setlur et al., 2024; Xia et al., 2025), model unlearning (Ji et al., 2024; Zhang et al., 2025), and time series (Cao et al., 2025; Ye et al., 2025). However, fine-tuning LLMs remains challenging for many practitioners because available hardware is often distributed across multiple commodity GPUs rather than concentrated in a single high-memory accelerator. In such environments, no individual GPU may have sufficient memory to accommodate the entire model and its training states, even when the aggregate memory across devices is sufficient. Although memory-efficient zeroth-order methods such as MeZO (Malladi et al., 2023) can fit models like OPT-6.7B on a sufficiently large modern GPU, such hardware is not universally available in practice. Many users instead rely on multiple smaller GPUs, making model parallelism (MP) necessary rather than merely convenient. MP, which partitions a model across multiple devices, is therefore a widely adopted solution for training and fine-tuning large models. Existing frameworks, including Megatron-LM (Shoeybi et al., 2019), DeepSpeed (Rasley et al., 2020), and MegaScale (Jiang et al., 2024), have demonstrated the effectiveness of this paradigm at scale. Most existing works focus on applying first-order (FO) method in the MP scenario. However, the combination of FO and MP inevitably introduce two substantial costs: (1) **High Communication Overhead:** FO methods (e.g., stochastic gradient descent) require frequent exchange of high-dimensional gradients and activations across nodes, making communication a major bottleneck in MP; (2) **High Memory Overhead:**

FO methods also demand storing gradients, optimizer states, and cached activations, further straining memory and often exceeding a single node's capacity. Scaling typically requires more GPU resources and aggressive parallelization, which amplifies communication and hardware overhead.

Recently, ZO methods have gained significant attention because they only require forward passes, offering substantial memory savings (Malladi et al., 2023; Zhang et al., 2024; Zhao et al., 2025). Yet, existing ZO research largely focuses on optimization theory, single-node or federated learning setups, but the potential of applying ZO optimization within a MP framework for fine-tuning LLMs remains unexplored. This gap naturally raises a key research question:

> *Q: Can we design a ZO MP framework simultaneously achieving high memory and communication efficiency while preserving comparable performance?*

To address this question, we propose `SparQ`, a ZO MP framework with split layer allocation informed by quantization-induced activation sparsity, aiming to simultaneously improve memory and communication efficiency while achieving comparable performance, compared to AQ-SGD. Our key contributions are summarized as follows:

- **Empirical Observation.** We empirically observe employing quantization to immediate outputs after commonly-used activation functions (e.g., ReLU, GELU, SwiGLU) in Transformer (Vaswani, 2017) architectures during LLM fine-tuning can result in pervasively sparse activations, with the proportion of nonzero entries across layers ranging between $1\% \sim 30\%$.

- **Framework Design.** Inspired by such inherent sparsity, we propose `SparQ`, a ZO model-parallel framework with **Sp**lit layer **a**llocation info**r**med by **Q**uantization-induced activation sparsity. It includes three key components: **(1) ZO optimization for gradient-free fine-tuning**: Prior studies have shown that ZO optimization is effective for LLM fine-tuning due to the presence of low-dimensional subspaces or low-rank Hessian landscapes (Malladi et al., 2023). We exploit its gradient-free nature to eliminate the need for storing and transmitting gradients in MP-based fine-tuning, significantly reducing both memory and communication overhead introduced by gradients. **(2) Quantization-induced high activation sparsity**: In Sec. 2.4, we observe that applying quantization to intermediate outputs after activation functions in Transformer architectures induces higher sparsity than unquantized activations during LLM fine-tuning. **(3) Split layer allocation guided by activation sparsity**: `SparQ` strategically places split layers at these naturally sparse regions. By transmitting activations in sparse representation (i.e., only transmit nonzero entries and their indices) across partitions, communication cost is substantially reduced while maintaining comparable model performance, compared to AQ-SGD.

- **Theoretical Analysis.** We prove that `SparQ` across multiple nodes achieves a sublinear convergence rate of $\mathcal{O}(\sqrt{d/T})$ for non-convex functions, matching the rate of centralized ZO stochastic gradient descent (ZO-SGD).

- **Empirical Validation.** We evaluate `SparQ`'s test accuracy, memory and communication costs on various LLM fine-tuning tasks and observe that: (1) `SparQ` achieves superior efficiency when memory and communication costs are jointly considered while maintaining comparable model performance, as shown in Fig. 1; (2) `SparQ` reduces communication cost by about $50\% \sim 70\%$, compared to FO-SGD and the state-of-the-art activation compression method AQ-SGD (Wang et al., 2022).

## 2 Formulations and Preliminaries

Given the loss function $f$, our goal is to minimize the following objective function

$$\min_{\boldsymbol{x} \in \mathbb{R}^d} f(\boldsymbol{x}) := \mathbb{E}_{\xi \sim \mathcal{D}} \left[ f(\xi; \boldsymbol{x}) \right],$$

where $\boldsymbol{x}$ is a $d$-dimensional model parameter, and $\xi$ is a data sample (model input) selected from the distribution $\mathcal{D}$. In the following subsections, we formulate two core components of `SparQ`: 1) model parallelism technique, and 2) ZO optimization's gradient estimation methods.

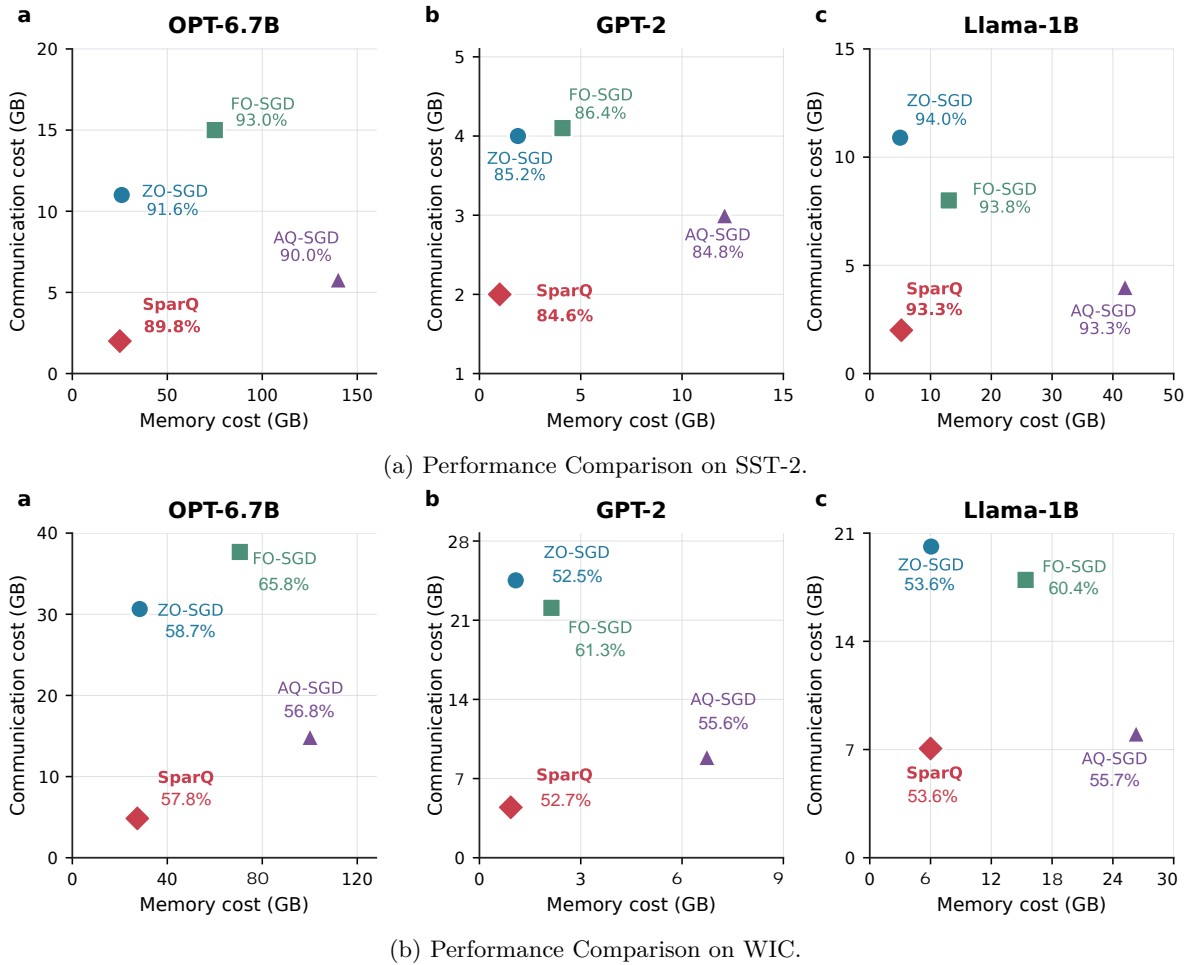

(a) Performance Comparison on SST-2.

(b) Performance Comparison on WIC.

Figure 1: Illustration of Memory Cost, Communication Cost, and Test Accuracy. Here we use SST-2 and WIC to fine-tune ReLU-based OPT, GELU-based GPT, and SwiGLU-based Llama models.

## 2.1 Model Parallelism Formulation

We begin with formulating model parallelism. In a model-parallel setting, the entire model is partitioned into $M$ parts, typically, with each computing node responsible for a distinct subset of the model's parameters. For clarity and brevity, we focus on the scenario in our main paper where the model is only divided into two submodels ($M = 2$), each residing on a separate node. This facilitates a concise presentation of the mathematical formulations, and an extension to an arbitrary number of partitions ($M > 2$) is provided in Appendix B.2.

The partition of model parameters denotes as $\boldsymbol{x} = \mathrm{col}[\boldsymbol{x}^{(1)}, \boldsymbol{x}^{(2)}]$, where $\boldsymbol{x} \in \mathbb{R}^d$, $\boldsymbol{x}^{(1)} \in \mathbb{R}^{d_1}$, $\boldsymbol{x}^{(2)} \in \mathbb{R}^{d_2}$, and $d_1 + d_2 = d$. Under this configuration, the loss function can be written as the composition of two functions:

$$f(\xi; \boldsymbol{x}) := F\left(S_2\big(S_1(\xi; \boldsymbol{x}^{(1)}); \boldsymbol{x}^{(2)}\big)\right), \tag{1}$$

where $F$ is the criterion function, $S_i$ represents a subset of network layers located on node $\mathcal{M}_i$ with model parameters $\boldsymbol{x}^{(i)}$. Specifically, for a given input $\xi$, $S_i(\xi; \boldsymbol{x}^{(i)})$ computes the forward activation on node $\mathcal{M}_i$, which is then transferred to the next node $\mathcal{M}_{i+1}$ where $S_2(\cdot; \boldsymbol{x}^{(2)})$ continues the computation. For notational simplicity, we also express the loss function using operator notation:

$$f(\xi; \boldsymbol{x}) = \left(F \circ S_2|_{\boldsymbol{x}^{(2)}} \circ S_1|_{\boldsymbol{x}^{(1)}}\right)(\xi), \tag{2}$$

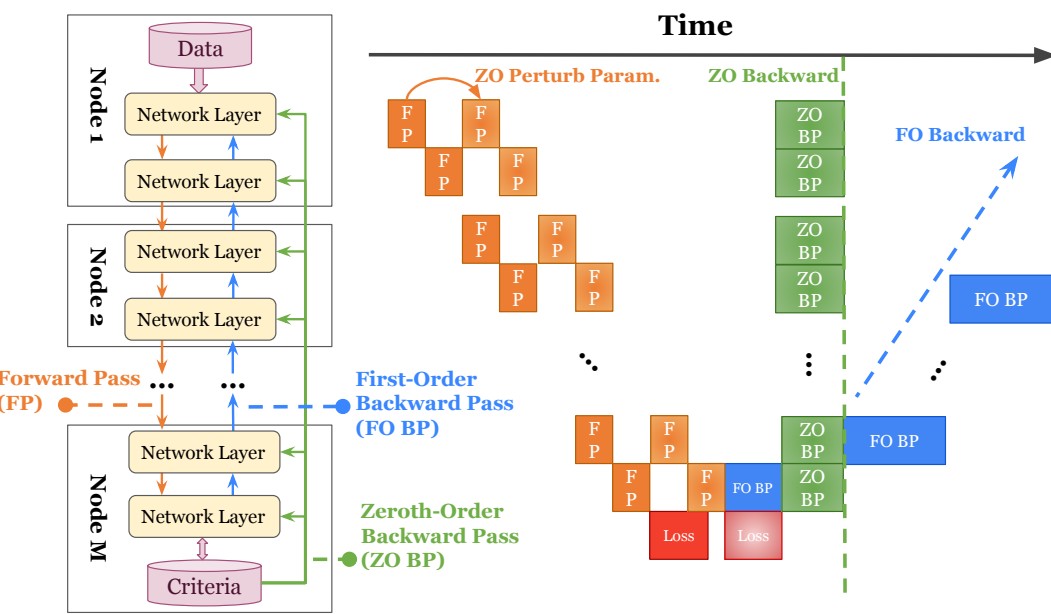

Figure 2: Execution Timeline Comparison of Zeroth-Order (e.g., `SparQ`) and First-Order Methods.

where $\circ$ denotes function composition (applying one function to the result of another). $S_1|_{\boldsymbol{x}^{(1)}}$ represents the first sub-model parameterized by $\boldsymbol{x}^{(1)}$ and acting on $\xi$. $S_2|_{\boldsymbol{x}^{(2)}}$ represents the second sub-model parameterized by $\boldsymbol{x}^{(2)}$ and acting on the output of $S_1|_{\boldsymbol{x}^{(1)}}$.

This two-node case ($M = 2$) can naturally be generalized to the multi-node case ($M > 2$) by further partitioning the model. Accordingly, the loss functions in (1) and (2) become

$$
\begin{aligned}
f(\xi; \boldsymbol{x}) =& F\left( S_M\left( \cdots \left( S_2\left( S_1(\xi; \boldsymbol{x}^{(1)}); \boldsymbol{x}^{(2)} \right); \cdots \right); \boldsymbol{x}^{(M)} \right) \right) \\
=& \left( F \circ S_M|_{\boldsymbol{x}^{(M)}} \circ \cdots \circ S_2|_{\boldsymbol{x}^{(2)}} \circ S_1|_{\boldsymbol{x}^{(1)}} \right)(\xi).
\end{aligned}
\tag{3}
$$

This formulation clearly delineates the flow of data across nodes in a framework, laying the groundwork for our subsequent discussions on optimization, memory and communication efficiency.

## 2.2 Zeroth-Order Optimization Formulation

Next, let us recap the fundamentals of ZO optimization, a gradient-free method that approximates the gradient by utilizing the finite difference method rather than explicit differentiation (Spall, 2002; Ghadimi & Lan, 2013; Nesterov & Spokoiny, 2017). Given a smoothing parameter $\mu > 0$, the number of perturbations $P$ and random perturbation vectors $\boldsymbol{u}$ drawn from a Gaussian distribution or a uniform ball, the ZO gradient estimate $\hat{G}$ can be computed as:

$$
\hat{G} = \frac{1}{P} \sum_{i=1}^{P} g_i \cdot \boldsymbol{u}_i = \frac{1}{P} \sum_{i=1}^{P} \frac{f(\xi; \boldsymbol{x} + \mu \boldsymbol{u}_i) - f(\xi; \boldsymbol{x})}{\mu} \cdot \boldsymbol{u}_i,
\tag{4}
$$

where Eq. (4) represents a biased forward difference approach. We distribute the content about unbiased central difference approach to Sec. B.1. These two approaches have been widely used to estimate the gradient in a ZO context (Ghadimi & Lan, 2013; Liu et al., 2020).

## 2.3 Model Parallelism Meets ZO Optimization

Having established the formulation for both ZO optimization and model parallelism, we now describe how to integrate ZO optimization into a model-parallel framework. For clarity, we still focus on the two-node

case ($M = 2$) here and defer the multi-node case ($M > 2$) to Appendix B.2. In this setup, node $\mathcal{M}_1$ is responsible solely for computing and transmitting the forward activations, while node $\mathcal{M}_2$ performs the gradient estimation, parameter updates and gradient scalar transmission.

In this work, we utilize classic Zeroth-Order Stochastic Gradient Descent (ZO-SGD) as our optimizer. Specifically, within the second submodel on $\mathcal{M}_2$, the ZO gradient scalar $g_t$ is computed using a forward difference method:

$$g_{i,t} = \frac{F(a_{i,t}^+; \boldsymbol{x}_t^{(2)} + \mu \boldsymbol{u}_{i,t}^{(2)}) - F(a_{i,t}; \boldsymbol{x}_t^{(2)})}{\mu}, \tag{5}$$

where $t$ is the iteration index, and $\boldsymbol{u}_{i,t}^{(2)}$ is the perturbation vector generated on node $\mathcal{M}_2$ during the $t$-th iteration. $a^+$ and $a$ are the activations obtained from the preceding sub-model on $\mathcal{M}_1$, defined as

$$a_{i,t}^+ = S_1\big(\xi_t\,;\, \boldsymbol{x}_t^{(1)} + \mu \boldsymbol{u}_{i,t}^{(1)}\big), \quad a_{i,t} = S_1\big(\xi_t\,;\, \boldsymbol{x}_t^{(1)}\big). \tag{6}$$

Then, the final ZO gradient estimate is given by

$$\hat{G}_t = \frac{1}{P} \sum_{i=1}^{P} g_{i,t} \cdot \boldsymbol{u}_{i,t}^{(2)}.$$

Integrating ZO optimization into a model-parallel framework offers a promising avenue to alleviate memory burdens associated with large-scale training. Unlike traditional FO methods that require storing gradients, ZO optimization relies exclusively on forward passes, thereby eliminating memory usage from storing gradients. However, this integration introduces a unique communication challenge. Since ZO optimization often depends on multiple perturbations to approximate gradients accurately, each computing node must transmit several forward activations across split layer per iteration. In contrast, FO methods generally exchange only a single forward activation and one backward gradient per iteration. Thus, while ZO optimization enables significant memory savings, its deployment in necessitates careful optimization of communication overhead to maintain training stability and performance. In Sec. 3, we will introduce how we address communication bottleneck.

### 2.4 Key Observations

We first summarize several key observations from Fig. 3 that motivate `SparQ`.

**(1) High Sparsity of Original ReLU-Based Activations in LLM Fine-Tuning.** Li et al. (2023) reported that ReLU-based activations exhibit high sparsity during LLM pre-training. Consistent with this finding, we observe a similar pattern in LLM fine-tuning in Fig. 3a: the proportion of nonzero values in nearly all ReLU-based activations remains below 10%. **(2) Low Sparsity of Original SwiGLU- and GELU-based Activations in LLM Fine-Tuning.** Smoother activation functions, such as SwiGLU and GELU, produce predominantly small but nonzero values, resulting in consistently dense activations with $99\% \sim 100\%$ nonzero entries. **(3) Quantization Induces High Sparsity across Various Activations.** Applying quantization can dramatically enhance activation sparsity during fine-tuning. Specifically, by using 4-bit quantization, ReLU-based activations become even sparser ( below 5% nonzero entries), SwiGLU-based activations stabilize below 20%, and nearly all GELU-based activations fall under 5% nonzero entries. This high sparsity motivates us to split the model immediately after the activation functions since activations can be efficiently encoded in sparse representation (described in Sec. 3.1), significantly reducing the communication burden associated with their transmission because we only need to transmit the nonzero values and their indices.

## 3 `SparQ` Framework Design

We introduce `SparQ`, a zeroth-order model-parallel framework with with **Sp**lit layer **a**llocation info**r**med by **Q**uantization-induced activation sparsity, reducing communication cost and minimizing memory costs for LLM fine-tuning in model parallelism. It employs ZO optimization to bypass the need for storing and transmitting

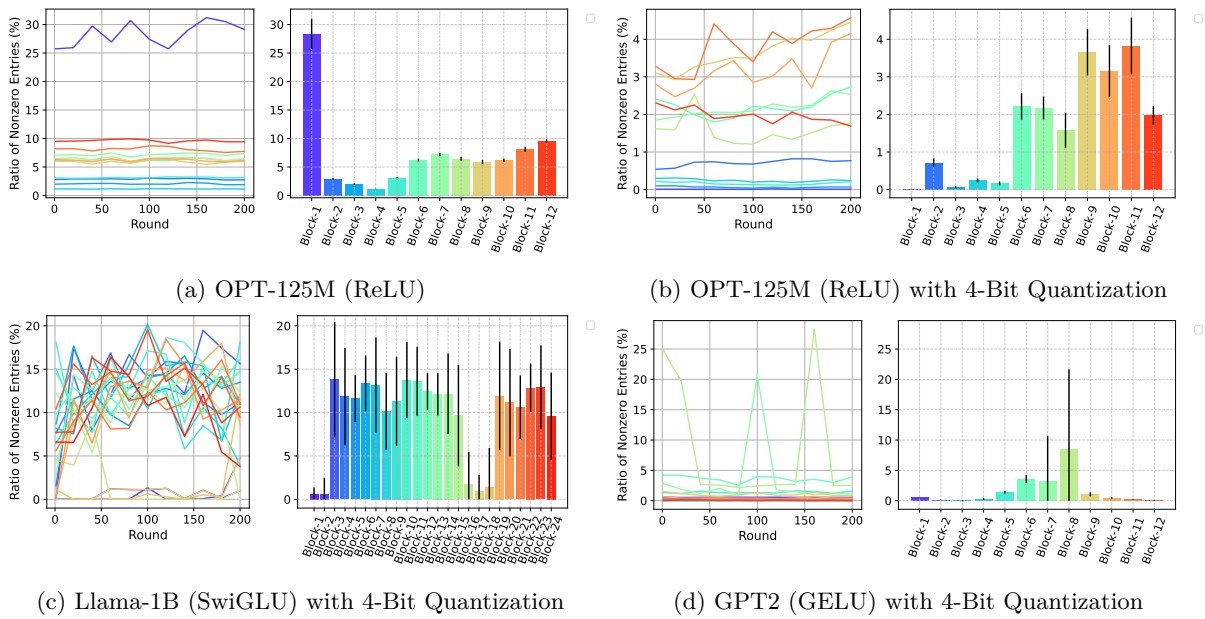

(a) OPT-125M (ReLU)

(b) OPT-125M (ReLU) with 4-Bit Quantization

(c) Llama-1B (SwiGLU) with 4-Bit Quantization

(d) GPT2 (GELU) with 4-Bit Quantization

Figure 3: Sparsity Levels of Various Activations Here we do not show the sparsity level of SwiGLU and GELU because they are very dense. Specifically, their ratio of nonzero entries is 99%-100%.

backward gradients. It is crucial to highlight our split strategy that is closely related to communication reduction. By capitalizing on the quantization-induced high sparsity of activations immediately after activation functions, we partition the model at these sparse areas to naturally lower communication cost.

In Alg. 1, we illustrate how `SparQ` operates on a two-GPU setup ($M = 2$) for brevity and distribute the multi-node case ($M > 2$) to Appendix B.2 due to the limited space. The entire model is partitioned into two segments, $S_1$ and $S_2$, which are deployed on two GPUs $\mathcal{M}_1$ and $\mathcal{M}_2$, respectively. The detailed procedure in the $t$-th iteration is as follows:

- **Step 1:** On node $\mathcal{M}_1$, the submodel $S_1$ computes activations $a_t^+$ and $a_t$ (Lines 5-7). Here, we perturb $\boldsymbol{x}_t^{(1)}$ element-wise following (Malladi et al., 2023) to reduce memory usage. By default, we use a biased forward difference approach in Alg. 1 and distribute the unbiased central difference approach to Appendix B.1. To lower communication cost, we design a compressor $\mathbb{C}$ that first applies quantization (e.g., 4 bit) to induce high sparsity in $a_t$ and $a_t^+$ and encodes them with sparse representations (i.e., only nonzero values and their indices). Subsequently, the compressed activations $\mathbb{C}(a_t)$ and $\mathbb{C}(a_t^+)$ are transmitted to the next node $\mathcal{M}_2$.

- **Step 2:** Upon receiving $\mathbb{C}(a_t)$ and $\mathbb{C}(a_t^+)$ from $\mathcal{M}_1$, node $\mathcal{M}_2$ feed them into submodel $S_2$ to compute the ZO gradient scalar $\hat{g}_t$ (Lines 11-15). The perturbation of $\boldsymbol{x}_t^{(2)}$ is performed in the same manner as in Step 1. Then, submodel parameters at $\mathcal{M}_2$ are updated via standard ZO-SGD (Line 16). Finally, only the scalar $\hat{g}_t$ is sent back to $\mathcal{M}_1$, instead of a high-dimensional first-order gradient, yielding substantial communication savings.

- **Step 3:** Once $\mathcal{M}_1$ receives $\hat{g}_t$, it updates sub-model parameters using the same ZO-SGD procedure applied on $\mathcal{M}_2$ (Line 19).

Here, we highlight the comparison of communication costs between central and forward difference methods. Given $P$ perturbations, in each round, utilizing the central difference method transmits $2P$ activations and $P$ gradient scalars, whereas utilizing the forward difference method transmits only $P + 1$ activations and $P$ gradient scalars.

In Fig. 2, we illustrate the execution timelines of our ZO method and conventional FO methods. The critical difference lies in the backward communication pattern: FO methods require layer-wise transmission

---

**Algorithm 1 SparQ** (forward difference method, $M = 2$)

---

1: **Initialize**: split model immediately after activation functions to get submodels $S_1$ and $S_2$, model parameter $\boldsymbol{x}_0 = \text{col}[\boldsymbol{x}_0^{(1)}, \boldsymbol{x}_0^{(2)}]$, learning rate $\eta$, smoothing parameter $\mu$, iterations $T$.

2: **for** $t = 0, 1, \cdots, T - 1$ **do**

3:      On node $\mathcal{M}_1$:

4:          Sample a random seed $s$ and a data sample $\xi_t$.

5:          Compute $a_t = S_1\big(\xi_t\,;\,\boldsymbol{x}_t^{(1)}\big)$

6:          $\boldsymbol{x}_t^{(1)} \leftarrow \text{Perturb}(\boldsymbol{x}_t^{(1)},\, \mu,\, s)$

7:          Compute $a_t^+ = S_1\big(\xi_t\,;\,\boldsymbol{x}_t^{(1)}\big)$

8:          $\boldsymbol{x}_t^{(1)} \leftarrow \text{Perturb}(\boldsymbol{x}_t^{(1)},\, -\mu,\, s)$

9:          Send $\mathbb{C}(a_t^+)$ and $\mathbb{C}(a_t)$, which are sparse representations of quantized $a_t^+$ and $a_t$, to $\mathcal{M}_2$.

10:      On node $\mathcal{M}_2$:

11:          Compute $f_t = F\big(S_2(\mathbb{C}(a_t))\,;\,\boldsymbol{x}_t^{(2)})\big)$

12:          $\boldsymbol{x}_t^{(2)} \leftarrow \text{Perturb}(\boldsymbol{x}_t^{(2)},\, \mu,\, s)$

13:          Compute $f_t^+ = F\big(S_2(\mathbb{C}(a_t^+))\,;\,\boldsymbol{x}_t^{(2)})\big)$

14:          $\boldsymbol{x}_t^{(2)} \leftarrow \text{Perturb}(\boldsymbol{x}_t^{(2)},\, -\mu,\, s)$

15:          $\hat{g}_t = \frac{1}{\mu}(f_t^+ - f_t)$

16:          $\boldsymbol{x}_{t+1}^{(2)} \leftarrow \text{Update}(\boldsymbol{x}_t^{(2)},\, \eta,\, \hat{g}_t,\, s)$

17:          Send $\hat{g}_t$ back to $\mathcal{M}_1$.          ▶ Only a scalar is transmitted in backward propagation

18:      On node $\mathcal{M}_1$:

19:          $\boldsymbol{x}_{t+1}^{(1)} \leftarrow \text{Update}(\boldsymbol{x}_t^{(1)},\, \eta,\, \hat{g}_t,\, s)$

20: **end for**

21: **Function** Perturb($\boldsymbol{x}$, $\mu$, $s$):

22:      Use random seed $s$ to reset random number generator

23:      **for** $x_i \in \boldsymbol{x}$ **do**

24:          $x_i \leftarrow x_i + \mu \cdot u$, where $u \sim \mathcal{N}(0, 1)$.

25:      **end for**

26:      **return** $x$

27: **Function** Update($\boldsymbol{x}$, $\eta$, $\hat{g}$, $s$):

28:      Use random seed $s$ to reset random number generator

29:      **for** $x_i \in \boldsymbol{x}$ **do**

30:          $x_i \leftarrow x_i - \eta \cdot \hat{g} \cdot u$, where $u \sim \mathcal{N}(0, 1)$          ▶ Standard Zeroth-Order SGD

31:      **end for**

32:      **return** $\boldsymbol{x}$

---

of high-dimensional gradients during backpropagation, incurring substantial communication and time costs. In contrast, our ZO method eliminates gradient tensors entirely and instead communicates only a single scalar per iteration, which can be broadcast to all nodes simultaneously. This design not only reduces the per-iteration communication complexity from dimension-dependent to dimension-free, but also eliminates sequential gradient exchanges, thereby reducing communication cost and shortening the overall execution timeline.

### 3.1 Why Activation Sparsity can be Helpful to Reduce Communication Cost?

Let the activation tensor contain $d$ entries and let $\rho$ denote the fraction of nonzero elements after quantization. Instead of transmitting the full dense activation, SparQ transmits only the nonzero values together with their indices. As a result, the amount of communicated data scales with $\rho d$ rather than $d$. Since the quantized activations observed in Sec. 2.4 are highly sparse across ReLU-, GELU-, and SwiGLU-based models, sparse

transmission can substantially reduce communication overhead while preserving the information required for downstream computation.

## 4 Theoretical Analysis

We start with all assumptions used in this work. Note that $f(\cdot)$ is the loss function, $\nabla f(\cdot)$ is the first-order gradient, $\hat{\nabla} f(\cdot)$ is the zeroth-order gradient. For simplicity, we only show two-node case in the following main paper and distribute the $M > 2$ case to Appendix C.3.

**Assumption 1 (Unbiased Function Estimation)** *We assume that the stochastic estimation of function $f$ is unbiased:* $\mathbb{E}\left[f(\xi_t; \boldsymbol{x})\right] = f(\boldsymbol{x})$.

**Assumption 2 (Unbiased Zeroth-Order Stochastic Gradients with Bounded Variance)** *The stochastic gradient $\hat{\nabla} f(\boldsymbol{x}_k; \xi_k)$ is unbiased, and its variance is bounded, so we have $\mathbb{E}[\hat{\nabla} f(\xi_k; \boldsymbol{x}_k)] = \nabla f(\boldsymbol{x}_k)$ and $\mathbb{E}\|\hat{\nabla} f(\xi_k; \boldsymbol{x}_k) - \nabla f(\boldsymbol{x}_k)\|^2 \le \sigma^2$.*

**Assumption 3 (Gradient Lipschitz Condition)** *For the loss function and each composition function, their gradients are Lipschitz-continuous such that for any $\xi_t$,*

$$\|\nabla f(\xi_t; \boldsymbol{x}) - \nabla f(\xi_t; \boldsymbol{y})\| \le L\|\boldsymbol{x} - \boldsymbol{y}\|$$

$$\left\|\nabla\big(S_1|_{\boldsymbol{x}^{(1)}}\big)(\xi_t) - \nabla\big(S_1|_{\boldsymbol{y}^{(1)}}\big)(\xi_t)\right\| \le L_1\|\boldsymbol{x}^{(1)} - \boldsymbol{y}^{(1)}\|$$

$$\left\|\nabla\big(F \circ S_2|_{\boldsymbol{x}^{(2)}}\big)(\zeta_t) - \nabla\big(F \circ S_2|_{\boldsymbol{y}^{(2)}}\big)(\zeta_t)\right\| \le L_2\|\boldsymbol{x}^{(2)} - \boldsymbol{y}^{(2)}\|$$

*Further, the gradients are bounded:* $\left\|\nabla\big(S_1|_{\boldsymbol{x}^{(1)}}\big)(\xi_t)\right\| \le C_{S_1}$ *and* $\left\|\nabla\big(F \circ S_2|_{\boldsymbol{x}^{(2)}}\big)(\zeta_t)\right\| \le C_{S_2}$.

**Assumption 4 (Bounded Output of Split Layer)** $\|S_1(\xi; \boldsymbol{x})\| \le L_{S_1}, \forall \boldsymbol{x}, \xi$.

**Assumption 5 (Compressor $\mathbb{C}(\cdot)$)** $\|\boldsymbol{x} - \mathbb{C}(\boldsymbol{x})\| \le \kappa\|\boldsymbol{x}\|$, *where* $\kappa \in [0, 1)$.

Under the assumptions above, we derive the lemma and obtain the `SparQ`'s nonconvex convergence bound the two-node case as follows.

**Lemma 1 (Distance between Gradients of Uncompressed and Compressed Activations)** *For Algorithm 1 with $M = 2$, the difference between gradients of uncompressed and compressed activations can be bounded as:* $\|\nabla f(\boldsymbol{x}_t) - \nabla \hat{f}(\boldsymbol{x}_t)\|^2 \le (1 + C_{S_1}^2)L_2^2\kappa^2 L_{S_1}^2$.

**Remark 1** *From Lemma 1, we observe that the upper bound is a constant, independent of learning rate $\eta$ and smoothing parameter $\mu$. This constant bound could, in principle, be further reduced. For example, AQ-SGD (Wang et al., 2022) applies a error-feedback technique on the compressed activation, but its memory cost is huge since it requires storing the previous information for each data sample on both nodes, which is mostly infeasible in practice. Our paper considers the memory limitation scenario, so we opt to compress the activation directly without any extra memory cost.*

**Theorem 1 (Convergence of `SparQ` under Non-Convexity, $M = 2$)** *Under the assumptions 1, 2, 3, 4 and 5, supposing that $\eta = \mathcal{O}(1/\sqrt{Td})$, $\mu \le 1/(d+6)\sqrt{T}$ and $D = f(\boldsymbol{x}_0) - f(\boldsymbol{x}^\star)$, then the sequence $\{\boldsymbol{x}_t\}$ generated by `SparQ` satisfies*

$$\frac{1}{T}\sum_{t=0}^{T-1} \mathbb{E}\|\nabla f(\boldsymbol{x}_t)\|^2 = \mathcal{O}\left(D\sqrt{d/T}\right) + \mathcal{O}\left(\sqrt{d/T}\sigma^2\right) + \mathcal{O}\left((1 + C_{S_1}^2)L_2^2\kappa^2 L_{S_1}^2\sqrt{d/T}\right). \tag{7}$$

**Remark 2** *The first term in the above bound is associated with the distance between the initial point and the optimal point. The second term is related to the variance of stochastic gradients. Both terms have the $O(\sqrt{d/T})$ convergence rate matching with the standard ZO method in the non-convex scenario. The third one is an extra term caused by the extra compression in the activation. It does not impact the overall convergence rate asymptotically. When using a lossless compressor ($\kappa = 0$) like sparse representation, then the third term disappears, and `SparQ` can achieve a convergence rate of $\mathcal{O}(\sqrt{d/T})$ under the non-convexity.*

## 5 Empirical Evaluation

### 5.1 Experiment Setup

**Models & Datasets**. We utilize three NLP datasets: SST2 (Socher et al., 2013) for sentiment classification, WIC (Pilehvar & Camacho-Collados, 2019) for context-sensitive word embeddings evaluation and RTE (Bowman et al., 2015) for textual entailment recognition. For each of them, we fine-tune OPT-125M, OPT-1.3B, OPT-6.7B (Zhang et al., 2022), GPT2 (Radford et al., 2019) and Llama-1B (Touvron et al., 2023) models and monitor their test accuracy, peak GPU memory usage and communication cost.

**Split Layer Selection.** In the two-node ($M = 2$) setting, to balance the computation and memory burden across the two nodes, we select the split layer near the middle block where the quantized output exhibits the highest sparsity. Accordingly, in our experiments, we split the OPT models immediately after the activation function in the middle block (e.g., Block 6 for OPT-125M), the Llama-1B model after the activation function in Block 15, and the GPT-2 model after the activation function in Block 7.

**Baselines.** To comprehensively evaluate the performance, we compare **SparQ** with several baselines: first-order SGD (FO-SGD), AQ-SGD (Wang et al., 2022) and ZO-SGD (i.e., MeZO (Malladi et al., 2023)). FO-SGD represents a centralized first-order method. AQ-SGD represents a first-order model parallel method with activation (using error feedback) and gradient compression. MeZO represents a recent memory-efficient ZO method without considering communication cost. For compression level, we use 4-bit quantization to compress activations for AQ-SGD and **SparQ**, and we employ 8-bit quantization to compress backward gradients for AQ-SGD.

**Ablation Experiments.** 1) splitting between blocks and splitting immediately after activation functions to explore the impact of split positions in Fig. 4; 2) using different quantization degrees (e.g., 1-bit, 2-bit, 4-bit and 8-bit) to investigate the influence of quantization levels in Table. 1.

### 5.2 Experiment Results

**Split Between Blocks v.s. Split Immediately After Activation Functions.** Fig. 4 shows our splitting strategy. As shown in (a)-(c), placing the split after activation functions leads to trainable models. Specifically, ReLU induces natural sparsity that is well-preserved after 4-bit quantization, while SwiGLU and GELU, though producing dense activations, still maintain sufficient information for effective training. In contrast, (d) shows that splitting between blocks severely damages the activation representation after quantization, making the model untrainable. These results show that our splitting strategy (a-c) effectively balances quantization and trainability, validating the design choice.

**SparQ's Performance with Different Quantization Levels.** Table 1 reports **SparQ**'s performance under four quantization schemes: 1-bit, 2-bit, 4-bit and 8-bit, under a single split setting ($M = 2$). We make two key observations: (1) stronger quantization (fewer bits) consistently reduces communication overhead; (2) accuracy degradation is relatively minor, even under aggressive quantization. It is worth noting that these results reflect the impact of quantization in the case of a single model split. When the model is split into multiple segments, the effect of quantization on both accuracy and communication overhead becomes more pronounced. Balancing accuracy and efficiency, we therefore select 4-bit quantization as the default configuration in subsequent experiments.

**Test Accuracy of SparQ with 4-Bit Quantization Matches ZO-SGD.** Table 2 shows that, compared to ZO-SGD, **SparQ** with 4-bit quantization attains competitive performance and lower communication overhead. Moreover, on several SST-2 and WIC tasks, **SparQ** even outperforms AQ-SGD.

**SparQ Achieves the Best Efficiency in Both Memory and Communication.** As reported in Table 2, **SparQ** achieves the lowest overhead when considering both memory and communication costs. On the memory side, compared with FO-SGD, **SparQ** reduces total peak usage by about $30 \sim 70\%$, while also avoiding the substantial extra per-sample storage required by AQ-SGD. This shows that **SparQ** can effectively scale to larger models where FO methods often encounter memory bottlenecks. On the communication side, **SparQ** stably outperforms both FO-SGD and AQ-SGD, with the forward-difference variant consistently achieving

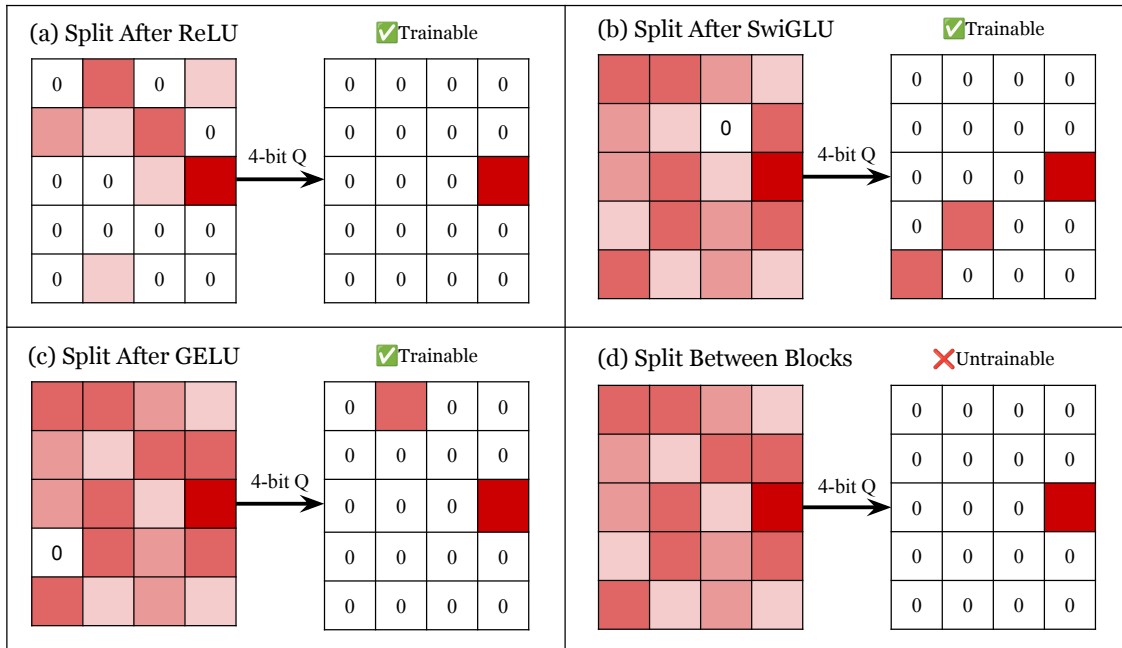

Figure 4: Activation Patterns Before and After 4-Bit Quantization at Different Split Positions. Each panel visualizes a representative activation pattern from a candidate split position during the forward pass, together with its corresponding 4-bit quantized version. Panels (a)-(c) show splits immediately after ReLU, SwiGLU, and GELU activation functions, while panel (d) shows a split between transformer blocks. A configuration is labeled "trainable" if it achieves meaningful test accuracy under the same fine-tuning setting, and "untrainable" if the test accuracy remains near random or near-zero performance throughout training. ReLU activations exhibit strong pre-quantization sparsity that is largely preserved after quantization. Although SwiGLU and GELU activations are denser, their quantized activations still retain sufficient informative nonzero elements for effective training. In contrast, splitting between blocks leads to severely distorted quantized activations, causing training to fail.

Table 1: Activation Quantization Levels' Impact on `SparQ`'s Test Accuracy and Total Communication Overhead. Hyper-parameter setup: batch size=32, $M = 2$. The reported communication cost is the total cumulative communication cost over the complete fine-tuning run.

| Model | Dataset | 1-bit Quantization | 2-bit Quantization | 4-bit Quantization | 8-bit Quantization |
|---|---|---|---|---|---|
| OPT-125M (ReLU) | SST-2 | 83.58%±0.10 (0.17 GB) | 84.34%±0.14 (0.23 GB) | 84.58%±0.11 (0.46 GB) | 84.63%±0.09 (0.92 GB) |
| | WIC | 52.01%±0.23 (0.52 GB) | 52.47%±0.20 (1.03 GB) | 53.42%±0.15 (1.24 GB) | 53.54%±0.19 (2.48 GB) |
| | RTE | 53.11%±0.19 (0.64 GB) | 53.29%±0.18 (1.28 GB) | 53.27%±0.20 (2.56 GB) | 54.79%±0.15 (5.12 GB) |
| OPT-1.3B (ReLU) | SST-2 | 89.84%±0.15 (0.31 GB) | 89.97%±0.17 (0.58 GB) | 92.34%±0.16 (1.20 GB) | 92.04%±0.22 (2.32 GB) |
| | WIC | 54.75%±0.16 (0.78 GB) | 55.50%±0.22 (1.57 GB) | 55.62%±0.19 (3.23 GB) | 56.07%±0.14 (6.26 GB) |
| | RTE | 57.03%±0.29 (2.49 GB) | 57.54%±0.31 (4.34 GB) | 57.13%±0.25 (7.18 GB) | 58.25%±0.21 (14.06 GB) |
| GPT2 (GELU) | SST-2 | 84.04%±0.22 (0.52 GB) | 84.15%±0.26 (0.91 GB) | 84.82%±0.13 (1.55 GB) | 85.12%±0.18 (2.64 GB) |
| | WIC | 51.59%±0.23 (1.18 GB) | 52.20%±0.20 (2.33 GB) | 52.70%±0.20 (3.79 GB) | 53.05%±0.16 (6.88 GB) |
| | RTE | 51.26%±0.19 (3.11 GB) | 52.01%±0.16 (5.48 GB) | 52.37%±0.15 (8.93 GB) | 52.78%±0.11 (14.72 GB) |
| Llama-1B (SwiGLU) | SST-2 | 89.32%±0.18 (0.65 GB) | 92.13%±0.20 (1.47 GB) | 93.44%±0.17 (2.42 GB) | 93.69%±0.16 (5.64 GB) |
| | WIC | 52.77%±0.21 (1.89 GB) | 53.35%±0.28 (3.53 GB) | 53.63%±0.19 (7.31 GB) | 53.82%±0.20 (10.22 GB) |
| | RTE | 53.62%±0.24 (3.47 GB) | 54.58%±0.21 (6.83 GB) | 55.21%±0.18 (12.46 GB) | 56.17%±0.17 (24.74 GB) |

the lowest communication cost across all model scales. On average, `SparQ` reduces communication overhead by more than 50% compared to AQ-SGD and over 70% compared to FO-SGD. Taken together, these results establish `SparQ` as the most efficient method overall, striking a favorable balance between maintaining accuracy and minimizing both memory and communication demands.

Table 2: Test Accuracy, Memory Cost, and Total Communication Cost. 1) Memory cost here means total peak memory usage. For AQ-SGD, the value before + is the total peak memory usage on two GPUs, and the value after + is the extra memory usage to store per-sample messages, which are on SSD or CPU, as Wang et al. (2022); 2) Key hyper-parameter setup: $P = 5$ for all ZO methods, batch size= 32; 3) "4Q" means 4-bit quantization. "forward" means forward difference method. 4) The reported communication cost is the total cumulative communication cost over the complete fine-tuning run.

| Model | Dataset | FO-SGD | AQ-SGD | ZO-SGD | Ours (4Q, forward) |
|---|---|---|---|---|---|
| OPT-125M | SST-2 | 87.50%±0.09 (2 GB, 2.8 GB) | 82.38%±0.17 (2+11 GB, 1.1 GB) | 85.21%±0.16 (1 GB, 3.1 GB) | 84.58%±0.10 (1 GB, **0.5 GB**) |
| | WIC | 54.07%±0.20 (3 GB, 7.2 GB) | 53.85%±0.17 (3+5 GB, 2.7 GB) | 53.58%±0.14 (2 GB, 8.3 GB) | 53.42%±0.15 (2 GB, **1.2 GB**) |
| | RTE | 56.53%±0.12 (11 GB, 15.7 GB) | 54.27%±0.17 (11+10 GB, 5.9 GB) | 53.46%±0.23 (6 GB, 17.1 GB) | 53.27%±0.20 (6 GB, **2.6 GB**) |
| OPT-1.3B | SST-2 | 91.66%±0.25 (15 GB, 8 GB) | 84.20%±0.21 (15+31 GB, 3 GB) | 90.62%±0.18 (6 GB, 8 GB) | 92.34%±0.16 (6 GB, **1 GB**) |
| | WIC | 63.56%±0.20 (16 GB, 19 GB) | 56.70%±0.18 (16+9 GB, 7 GB) | 55.77%±0.14 (7 GB, 21 GB) | 55.62%±0.19 (7 GB, **3 GB**) |
| | RTE | 70.83%±0.19 (41 GB, 42 GB) | 60.24%±0.18 (41+34 GB, 16 GB) | 57.32%±0.16 (11 GB, 47 GB) | 57.13%±0.25 (11 GB, **7 GB**) |
| OPT-6.7B | SST-2 | 94.44%±0.24 (76 GB, 15 GB) | 89.61%±0.18 (76+61 GB, 6 GB) | 92.13%±0.21 (26 GB, 11 GB) | 92.04%±0.24 (26 GB, **2 GB**) |
| | WIC | 65.78%±0.16 (78 GB, 39 GB) | 56.80%±0.24 (78+19 GB, 15 GB) | 58.66%±0.17 (27 GB, 31 GB) | 57.82%±0.23 (27 GB, **5 GB**) |
| | RTE | 71.12%±0.15 (202 GB, 83 GB) | 64.25%±0.18 (202+69 GB, 31 GB) | 63.52%±0.17 (29 GB, 63 GB) | 63.14%±0.20 (29 GB, **9 GB**) |
| GPT2 | SST-2 | 88.12%±0.20 (2 GB, 4 GB) | 84.47%±0.19 (2+10 GB, 3 GB) | 84.88%±0.14 (1 GB, 4 GB) | 84.82%±0.13 (1 GB, **2 GB**) |
| | WIC | 61.34%±0.18 (2 GB, 22 GB) | 55.56%±0.15 (2+5 GB, 9 GB) | 52.53%±0.21 (1 GB, 25 GB) | 52.70%±0.20 (1 GB, **4 GB**) |
| | RTE | 63.12%±0.14 (5 GB, 46 GB) | 55.86%±0.18 (5+10 GB, 20 GB) | 52.30%±0.10 (3 GB, 52 GB) | 52.37%±0.15 (3 GB, **9 GB**) |
| Llama-1B | SST-2 | 94.31%±0.20 (14 GB, 8 GB) | 93.23%±0.19 (14+30 GB, 4 GB) | 93.67%±0.15 (5 GB, 11 GB) | 93.44%±0.17 (5 GB, **2 GB**) |
| | WIC | 60.39%±0.15 (16 GB, 18 GB) | 55.66%±0.28 (16+10 GB, 8 GB) | 53.60%±0.20 (6 GB, 20 GB) | 53.63%±0.19 (6 GB, **7 GB**) |
| | RTE | 64.73%±0.17 (25 GB, 41 GB) | 59.05%±0.21 (25+30 GB, 17 GB) | 55.75%±0.13 (8 GB, 42 GB) | 55.21%±0.18 (8 GB, **12 GB**) |

# 6 Related Work

**Model Parallelism (MP).** MP (Dean et al., 2012) is a fundamental technique in distributed deep learning that partitions a deep neural network into disjoint segments, each assigned to a separate computing node (e.g., a GPU or machine). Building on MP, various algorithms have been proposed. Among these, AQ-SGD (Wang et al., 2022) is particularly relevant to this work. It employs pipeline parallelism and can function effectively over slow networks but introduces a huge extra memory cost.

**Zeroth-Order (ZO) Optimization.** ZO optimization is a gradient-free approach that estimates gradients using only differences in function values and random perturbation vectors (Liu et al., 2020), in contrast to first-order methods, which rely on explicitly computed gradients. Prior research has demonstrated the efficacy of ZO optimization in black-box attack (Kariyappa et al., 2021; Yu et al., 2024), reinforcement learning (Pan et al., 2022; Jing et al., 2024), communication savings (Fang et al., 2022; Qin et al., 2024; Li et al., 2025; 2026a; Liang et al., 2025), LLM fine-tuning (Li et al., 2026b), etc. In addition, ZO optimization has been shown to lower memory consumption during LLM fine-tuning. For example, MeZO (Malladi et al., 2023) and LOZO (Chen et al., 2025) utilize ZO optimization to perform solely forward passes, thereby eliminating the need to store gradients from backward propagation. Yet, the integration of ZO optimization within MP frameworks remains largely unexplored. Moreover, recent efforts have explored parameter-sparse ZO fine-tuning. For example, Sparse MeZO (Liu et al., 2025) and SensZOQ (Guo et al., 2025) sparsify the parameter space by limiting the set of perturbed or updated weights, thereby improving the efficiency and statistical quality of ZO gradient estimation in centralized training. Our work instead exploits activation sparsity in model-parallel settings, where communication among partitions is a key bottleneck. Specifically, we reduce the amount of transmitted intermediate activations rather than the number of optimized parameters. As a result, the two lines of work target orthogonal sources of efficiency (parameter sparsity versus activation sparsity) and are developed for different system settings.

**Activation Compression & Sparsity.** Compression is widely adopted to reduce communication and memory overhead in distributed systems. Popular compression techniques, such as quantization (Gray & Neuhoff, 1998; Alistarh et al., 2017; Horváth et al., 2023; Lin et al., 2024) and sparsification (Wangni et al., 2018; Yang et al., 2021; Yoon & Oh, 2023), have primarily targeted gradients and weights. Recently, attention has shifted toward compressing activations (Evans & Aamodt, 2021; Liu et al., 2021; 2022; Chen et al., 2021; Eliassen & Selvan, 2024; Lin et al., 2024). For example, ActNN (Chen et al., 2021) and ALAM (Woo et al., 2024) quantize activations to improve memory efficiency. Moreover, AQ-SGD (Wang et al., 2022) quantizes backward gradients and the changes of forward activation to enable communication-efficient training in

pipeline parallel architectures. However, its reliance on error-feedback mechanism to guarantee convergence necessitates storing per-sample information on CPUs or SSDs, thereby imposing a significant extra memory burden.

## 7 Conclusion

In summary, we introduce `SparQ`, a ZO model-parallel framework with split layer allocation informed by quantization-induced activation sparsity, reducing communication overhead and minimizes memory costs for LLM fine-tuning under MP. Our theoretical analysis establishes a sublinear convergence rate in non-convex settings, and empirical results show that `SparQ` reduces GPU memory consumption by more than $3\times$ and communication cost by over 50% compared to state-of-the-art baselines. These findings highlight the potential of ZO, sparsity-guided frameworks for scaling LLM fine-tuning, and open up promising directions for future research on distributed optimization.

### Broader Impact Statement

This paper focuses on reducing communication and memory overhead in model parallelism via zeroth-order optimization methods. We do not believe that there are specific negative societal consequences that must be highlighted here.

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

# Appendix

# A    Additional Experiment Details and Results

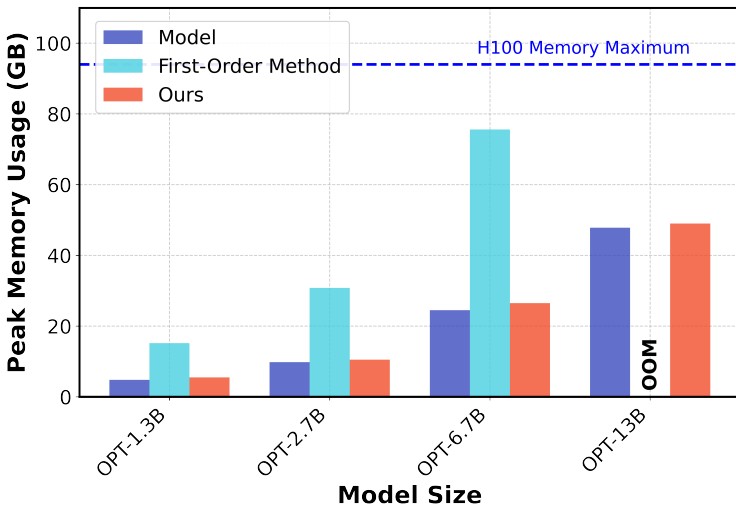

Figure 5: GPU Peak Memory Usage across Multiple LLMs on SST-2 Dataset. "OOM" means out of memory. Experiment setup: train batch size=test batch size= 32, momentum= 0.

## A.1    GPU Peak Memory Usage

We execute a group of experiments to test GPU peak memory usages by fine-tuning the SST2 dataset across OPT-1.3B, OPT-2.7B, OPT-6.7B and OPT-13B models. Except for different model sizes, all hyperparameter setups are the same. Fig. 5 reveals several key empirical observations:

1) The memory overhead for first-order methods is over $3\times$ more than that for memory-efficient zeroth-order methods (e.g., MeZO and `SparQ`).

2) The memory cost for `SparQ` is approximately equal to the model size, matching the conclusion in MeZO paper (Malladi et al., 2023) and demonstrating the significant memory reduction compared with first-order methods.

# B    `SparQ`'s Technical Details and Framework Extensions

## B.1    `SparQ` with Central Difference Method

In our main paper, we focus on the biased forward difference method to estimate zeroth-order gradients because of its advantage of less activation transmission and comparable precision performance. Here, we introduce another widely used gradient estimation approach - the central difference method, which estimates the ZO gradient as:

$$\hat{G} = \frac{1}{P} \sum_{i=1}^{P} g_i \cdot \boldsymbol{u}_i = \frac{1}{P} \sum_{i=1}^{P} \frac{f(\xi; \boldsymbol{x} + \mu\boldsymbol{u}_i) - f(\xi; \boldsymbol{x} - \mu\boldsymbol{u}_i)}{2\mu} \cdot \boldsymbol{u}_i, \tag{8}$$

Additionally, when integrating ZOO using the unbiased central difference method into model parallelism formulation, the expression in Eq. (5) of computing the zeroth-order gradient scalar will be replaced by

$$g_{i,t} = \frac{F(a_{i,t}^{+}; \boldsymbol{x}_t^{(2)} + \mu\boldsymbol{u}_{i,t}^{(2)}) - F(a_{i,t}^{-}; \boldsymbol{x}_t^{(2)} - \mu\boldsymbol{u}_{i,t}^{(2)})}{2\mu}, \tag{9}$$

where activations are computed by

$$a_{i,t}^+ = S_1\big(\xi_t\,;\,\boldsymbol{x}_t^{(1)} + \mu\boldsymbol{u}_{i,t}^{(1)}\big), \;\; a_{i,t}^- = S_1\big(\xi_t\,;\,\boldsymbol{x}_t^{(1)} - \mu\boldsymbol{u}_{i,t}^{(1)}\big). \tag{10}$$

In our experiments, the main performance difference of forward and central difference methods lies in communication overhead. Overall, central difference incurs higher communication cost compared to forward difference, as analyzed in Sec. B.2.

### B.2 `SparQ` Across Multiple Computing Nodes ($M > 2$)

Our framework can also be straightforwardly extended to multiple computing node scenarios ($M > 2$). Alg. 2 demonstrates `SparQ` using the central difference method under the multi-node case ($M > 2$), and Alg. 3 shows `SparQ` using the forward difference method under the multi-node case ($M > 2$). In Sec. C.3, we provide the convergence theorem and its proof of the $M > 2$ case. The main impact of LLM fine-tuning on multiple computing nodes is the increased communication overhead attributable to the additional split layers. Nevertheless, compared with AQ-SGD (Wang et al., 2022), `SparQ` still can achieving $1 \sim 2\times$ communication savings.

**Communication Cost Analysis ($M > 2$).** Given the number of computing nodes $M$, and the number of perturbations $P$, the central difference method requires transmitting $2 \times P \times (M - 1)$ activations and $P \times (M - 1)$ gradient scalars per training round, whereas the forward difference method requires transmitting only $(P + 1) \times (M - 1)$ activations and $P \times (M - 1)$ gradient scalars per round. Therefore, in general, the communication overhead of using the forward difference method is lower than the cost of using the central difference method.

## C  Proof

### C.1  Lemmas

The following Lemma 2 and Lemma 3 are relative to zeroth-order optimization and have been commonly used in zeroth-order proof. It is worth pointing out that Lemma 3 is dependent on the forward difference method, which can be known by (14). Consequently, we use it to prove the convergence of Alg. 1 ($M = 2$) and Alg. 3 ($M > 2$).

**Lemma 2** $f \in \mathcal{C}_L^{1,1}(\mathbb{R}^d)$ *if $f$ is differentiable and satisfies*

$$\|\nabla f(\boldsymbol{x}) - \nabla f(\boldsymbol{y})\| \le L\|\boldsymbol{x} - \boldsymbol{y}\|. \tag{11}$$

*Also, we have*

$$|f(\boldsymbol{y}) - f(\boldsymbol{x}) - \langle \nabla f(\boldsymbol{x}), \boldsymbol{y} - \boldsymbol{x}\rangle| \le \frac{L}{2}\|\boldsymbol{y} - \boldsymbol{x}\|^2. \tag{12}$$

**Lemma 3** *(Ghadimi & Lan, 2013; Nesterov & Spokoiny, 2017) We define a smooth approximation of objective function $f$ as $f_\mu(\cdot)$ that can be formulated as*

$$f_\mu(\boldsymbol{x}) := \frac{1}{(2\pi)^{\frac{d}{2}}} \int f(\boldsymbol{x} + \mu\boldsymbol{u})e^{-\frac{1}{2}\|\boldsymbol{u}\|^2} d\boldsymbol{u} = \mathbb{E}\left[f(\boldsymbol{x} + \mu\boldsymbol{u})\right] \tag{13}$$

*Then, for any $f \in \mathcal{C}_L^{1,1}$, the following statements hold.*

*(a) The gradient of $f_\mu(\cdot)$ is $L_\mu$-Lipschitz continuous where $L_\mu \le L$. $\nabla f_\mu(\boldsymbol{x})$ can be shown as*

$$\nabla f_\mu(\boldsymbol{x}) = \frac{1}{(2\pi)^{\frac{d}{2}}} \int \frac{f(\boldsymbol{x} + \mu\boldsymbol{u}) - f(\boldsymbol{x})}{\mu}\boldsymbol{u}e^{-\frac{1}{2}\|\boldsymbol{u}\|^2} d\boldsymbol{u}. \tag{14}$$

---

**Algorithm 2 SparQ** ($M > 2$) with Central Difference Method

---

1: **Initialize**: split model immediately after activation functions to get submodels $S_1$, ..., $S_M$, model parameter $\mathbf{x}_0 = \text{col}[\boldsymbol{x}_0^{(1)}, \cdots, \mathbf{x}_0^{(M)}]$, learning rate $\eta$, smoothing parameter $\mu$, the number of iterations $T$.

2: **for** $t = 0, 1, ..., T - 1$ **do**

3:     On node $\mathcal{M}_1$:

4:         Sample a random seed $s$ and a data sample $\xi_t$

5:         $\boldsymbol{x}_t^{(1)} \leftarrow \text{Perturb}(\boldsymbol{x}_t^{(1)}, \mu, s)$

6:         Compute $a_t^+ = S_1\big(\xi_t ; \boldsymbol{x}_t^{(1)}\big)$

7:         $\boldsymbol{x}_t^{(1)} \leftarrow \text{Perturb}(\boldsymbol{x}_t^{(1)}, -2\mu, s)$

8:         Compute $a_t^- = S_1\big(\xi_t ; \boldsymbol{x}_t^{(1)}\big)$

9:         $\boldsymbol{x}_t^{(1)} \leftarrow \text{Perturb}(\boldsymbol{x}_t^{(1)}, \mu, s)$

10:        Send $\mathbb{C}(a_t^+)$ and $\mathbb{C}(a_t^-)$, which are sparse representations of quantized $a_t^+$ and $a_t^-$, to $\mathcal{M}_2$.

11:       Here we skip the description of all processes on $\mathcal{M}_2 \cdots \mathcal{M}_{M-1}$ because they have the similar process.

12:     On node $\mathcal{M}_M$:

13:         $\boldsymbol{x}_t^{(M)} \leftarrow \text{Perturb}(\boldsymbol{x}_t^{(M)}, \mu, s)$

14:         Compute $f_t^+ = F\big(S_M(\mathbb{C}(a_t^+) ; \boldsymbol{x}_t^{(M)})\big)$

15:         $\boldsymbol{x}_t^{(M)} \leftarrow \text{Perturb}(\boldsymbol{x}_t^{(M)}, -2\mu, s)$

16:         Compute $f_t^- = F\big(S_M(\mathbb{C}(a_t^-) ; \boldsymbol{x}_t^{(M)})\big)$

17:         $\boldsymbol{x}_t^{(M)} \leftarrow \text{Perturb}(\boldsymbol{x}_t^{(M)}, \mu, s)$

18:         $\hat{g}_t = \frac{1}{2\mu}(f_t^+ - f_t^-)$

19:         $\boldsymbol{x}_{t+1}^{(M)} \leftarrow \text{Update}(\boldsymbol{x}_t^{(M)}, \eta, \hat{g}_t, s)$

20:         Send $\hat{g}_t$ back to $\mathcal{M}_i$, where $i \in [1, \cdots, M - 1]$.       ▶ Only a scalar is transmitted

21:     On nodes $\mathcal{M}_i$, $i \in [1, ..., M - 1]$:

22:         $\boldsymbol{x}_{t+1}^{(i)} \leftarrow \text{Update}(\boldsymbol{x}_t^{(i)}, \eta, \hat{g}_t, s)$

23: **end for**

24: **Function** Perturb($\boldsymbol{x}$, $\mu$, $s$):

25:     Use random seed $s$ to reset random number generator

26:     **for** $x_i \in \boldsymbol{x}$ **do**

27:       $x_i \leftarrow x_i + \mu \cdot u$, where $u \sim \mathcal{N}(0, 1)$.

28:     **end for**

29:     **return** $x$

30: **Function** Update($\boldsymbol{x}$, $\eta$, $\hat{g}$, $s$):

31:     Use random seed $s$ to reset random number generator

32:     **for** $x_i \in \boldsymbol{x}$ **do**

33:       $x_i \leftarrow x_i - \eta \cdot \hat{g} \cdot u$, where $u \sim \mathcal{N}(0, 1)$.       ▶ Standard Zeroth-Order SGD

34:     **end for**

35:     **return** $\boldsymbol{x}$

---

*(b) For any $\boldsymbol{x} \in \mathbb{R}^n$,*

$$|f_\mu(\boldsymbol{x}) - f(\boldsymbol{x})| \leq \frac{\mu^2}{2} Ld \tag{15}$$

$$\|\nabla f_\mu(\boldsymbol{x}) - \nabla f(\boldsymbol{x})\| \leq \frac{\mu}{2} L(d + 3)^{\frac{3}{2}} \tag{16}$$

*(c) For any $\boldsymbol{x} \in \mathbb{R}^n$,*

$$\frac{1}{\mu^2} \mathbb{E}_{\boldsymbol{u}} \left[ \Big(f(\boldsymbol{x} + \mu\boldsymbol{u}) - f(\boldsymbol{x})\Big)^2 \|\boldsymbol{u}\|^2 \right] \leq \frac{\mu^2}{2} L^2 (d + 6)^3 + 2(d + 4)\|\nabla f(\boldsymbol{x})\|^2 \tag{17}$$

---

**Algorithm 3 SparQ** ($M > 2$) with Forward Difference Method

---

1: **Initialize**: split model immediately after activation functions to get submodels $S_1$, ..., $S_M$, model parameter $\mathbf{x}_0 = \mathrm{col}[\boldsymbol{x}_0^{(1)}, \cdots, \mathbf{x}_0^{(M)}]$, learning rate $\eta$, smoothing parameter $\mu$, the number of iterations $T$.
2: **for** $t = 0, 1, ..., T-1$ **do**
3:     On node $\mathcal{M}_1$:
4:         Sample a random seed $s$ and a data sample $\xi_t$
5:         Compute $a_t = S_1\big(\xi_t\,;\,\boldsymbol{x}_t^{(1)}\big)$
6:         $\boldsymbol{x}_t^{(1)} \leftarrow \text{Perturb}(\boldsymbol{x}_t^{(1)}, \mu, s)$
7:         Compute $a_t^+ = S_1\big(\xi_t\,;\,\boldsymbol{x}_t^{(1)}\big)$
8:         $\boldsymbol{x}_t^{(1)} \leftarrow \text{Perturb}(\boldsymbol{x}_t^{(1)}, -\mu, s)$
9:         Send $\mathbb{C}(a_t^+)$ and $\mathbb{C}(a_t)$, which are sparse representations of quantized $a_t^+$ and $a_t$, to $\mathcal{M}_2$.
10:        Here we skip the description of all processes on $\mathcal{M}_2 \cdots \mathcal{M}_{M-1}$ because they have the similar process.

11:     On node $\mathcal{M}_M$:
12:         Compute $f_t = F\big(S_M(\mathbb{C}(a_t)\,;\,\boldsymbol{x}_t^{(M)})\big)$
13:         $\boldsymbol{x}_t^{(M)} \leftarrow \text{Perturb}(\boldsymbol{x}_t^{(M)}, \mu, s)$
14:         Compute $f_t^+ = F\big(S_M(\mathbb{C}(a_t^+)\,;\,\boldsymbol{x}_t^{(M)})\big)$
15:         $\boldsymbol{x}_t^{(M)} \leftarrow \text{Perturb}(\boldsymbol{x}_t^{(M)}, -\mu, s)$
16:         $\hat{g}_t = \frac{1}{\mu}(f_t^+ - f_t)$
17:         $\boldsymbol{x}_{t+1}^{(M)} \leftarrow \text{Update}(\boldsymbol{x}_t^{(M)}, \eta, \hat{g}_t, s)$
18:         Send $\hat{g}_t$ back to $\mathcal{M}_i$, where $i \in [1, \cdots, M-1]$.         ▶ Only a scalar is transmitted
19:     On nodes $\mathcal{M}_i$, $i \in [1, ..., M-1]$:
20:         $\boldsymbol{x}_{t+1}^{(i)} \leftarrow \text{Update}(\boldsymbol{x}_t^{(i)}, \eta, \hat{g}_t, s)$
21: **end for**

22: **Function** Perturb($\boldsymbol{x}$, $\mu$, $s$):
23:     Use random seed $s$ to reset random number generator
24:     **for** $x_i \in \boldsymbol{x}$ **do**
25:       $x_i \leftarrow x_i + \mu \cdot u$, where $u \sim \mathcal{N}(0,1)$.
26:     **end for**
27:     **return** $x$

28: **Function** Update($\boldsymbol{x}$, $\eta$, $\hat{g}$, $s$):
29:     Use random seed $s$ to reset random number generator
30:     **for** $x_i \in \boldsymbol{x}$ **do**
31:       $x_i \leftarrow x_i - \eta \cdot \hat{g} \cdot u$, where $u \sim \mathcal{N}(0,1)$.         ▶ Standard Zeroth-Order SGD
32:     **end for**
33:     **return** $\boldsymbol{x}$

---

Following form (16) and utilizing Jensen's inequality $\|a\|^2 \leq 2\|a-b\|^2 + 2\|b\|^2$, we have

$$\|\nabla f_\mu(\boldsymbol{x})\|^2 \leq 2\|\nabla f(\boldsymbol{x})\|^2 + \frac{\mu^2}{2}L^2(d+3)^3, \tag{18}$$

$$\|\nabla f(\boldsymbol{x})\|^2 \leq 2\|\nabla f_\mu(\boldsymbol{x})\|^2 + \frac{\mu^2}{2}L^2(d+3)^3. \tag{19}$$

Moreover, we denote $f_\mu^* := \min_{\boldsymbol{x} \in \mathbb{R}^d} f_\mu(\boldsymbol{x})$ and conclude $|f_\mu^* - f^*| \leq \frac{\mu^2 Ld}{2}$ from (15). Then, we further conclude that

$$-\mu^2 Ld \leq \big[f_\mu(\boldsymbol{x}) - f_\mu^*\big] - \big[f(\boldsymbol{x}) - f^*\big] \leq \mu^2 Ld \tag{20}$$

**Lemma 4 (Distance between Gradients of Uncompressed and Compressed Activations, $M = 2$)**
*For Alg. 1, the difference between gradients of uncompressed and compressed activations can be bounded by a constant term as follows:*

$$\left\| \nabla f(\boldsymbol{x}_t) - \nabla \hat{f}(\boldsymbol{x}_t) \right\|^2 \leq (1 + C_{S_1}^2) L_2^2 \kappa^2 L_{S_1}^2.$$

*Proof of Lemma 4:*

$$
\begin{aligned}
\left\| \nabla f(\boldsymbol{x}_t) - \nabla \hat{f}(\boldsymbol{x}_t) \right\|^2 =& \| \nabla_{\boldsymbol{x}^{(1)}} f(\boldsymbol{x}_t) - \nabla_{\boldsymbol{x}^{(1)}} \hat{f}(\boldsymbol{x}_t) \|^2 + \| \nabla_{\boldsymbol{x}^{(2)}} f(\boldsymbol{x}_t) - \nabla_{\boldsymbol{x}^{(2)}} \hat{f}(\boldsymbol{x}_t) \|^2 \\
=& \left\| \nabla_{\boldsymbol{x}^{(1)}} (F \circ S_2 \circ S_1) \big|_{(\boldsymbol{x}_t^{(1)}, \boldsymbol{x}_t^{(2)})} - \nabla_{S_1} (F \circ S_2) \big|_{(\mathbb{C}(S_1(\xi; \boldsymbol{x}_t^{(1)})), \boldsymbol{x}_t^{(2)})} \cdot \nabla_{\boldsymbol{x}^{(1)}} S_1 \big|_{\boldsymbol{x}_t^{(1)}} \right\|^2 \\
& + \left\| \nabla_{\boldsymbol{x}^{(2)}} (F \circ S_2) \big|_{(S_1(\xi; \boldsymbol{x}_t^{(1)}), \boldsymbol{x}_t^{(2)})} - \nabla_{\boldsymbol{x}^{(2)}} (F \circ S_2) \big|_{(\mathbb{C}(S_1(\xi; \boldsymbol{x}_t^{(1)})), \boldsymbol{x}_t^{(2)})} \right\|^2 \\
\leq& C_{S_1}^2 L_2^2 \left\| (\mathbb{C}(S_1(\xi, \boldsymbol{x}_t^{(1)})), \boldsymbol{x}_t^{(2)}) - (S_1(\xi, \boldsymbol{x}_t^{(1)}), \boldsymbol{x}_t^{(2)}) \right\|^2 \\
& + L_2^2 \left\| (\mathbb{C}(S_1(\xi; \boldsymbol{x}_t^{(1)})); \boldsymbol{x}_t^{(2)}) - (S_1(\xi; \boldsymbol{x}_t^{(1)}); \boldsymbol{x}_t^{(2)}) \right\|^2 \\
=& (1 + C_{S_1}^2) L_2^2 \left\| (\mathbb{C}(S_1(\xi; \boldsymbol{x}_t^{(1)})); \boldsymbol{x}_t^{(2)}) - (S_1(\xi; \boldsymbol{x}_t^{(1)}); \boldsymbol{x}_t^{(2)}) \right\|^2 \\
\leq& (1 + C_{S_1}^2) L_2^2 \kappa^2 \left\| S_1(\xi; \boldsymbol{x}_t^{(1)}) \right\|^2 \\
\leq& (1 + C_{S_1}^2) L_2^2 \kappa^2 L_{S_1}^2, \quad\quad\quad\quad\quad\quad\quad\quad\quad (21)
\end{aligned}
$$

where we get (21) by assumption 4.  ∎

## C.2  Proof of Theorem 1

Before showing the proof of the theorem, we present notations, definitions and update rules used in the following proof.

$$
\boldsymbol{x}_{t+1}^{(1)} = \boldsymbol{x}_t^{(1)} - \eta \cdot \hat{G}_\mu(\boldsymbol{x}_t, \xi_t, \boldsymbol{u}_t) = \boldsymbol{x}_t^{(1)} - \eta \cdot \hat{g}_\mu(\boldsymbol{x}_t, \xi_t, \boldsymbol{u}_t) \cdot \boldsymbol{u}_t^{(1)} \quad\quad (22)
$$
$$
\boldsymbol{x}_{t+1}^{(2)} = \boldsymbol{x}_t^{(2)} - \eta \cdot \hat{G}_\mu(\boldsymbol{x}_t, \xi_t, \boldsymbol{u}_t) = \boldsymbol{x}_t^{(2)} - \eta \cdot \hat{g}_\mu(\boldsymbol{x}_t, \xi_t, \boldsymbol{u}_t) \cdot \boldsymbol{u}_t^{(2)} \quad\quad (23)
$$

Introducing $\boldsymbol{x}_t = \text{col}[\boldsymbol{x}_t^{(1)}, \boldsymbol{x}_t^{(2)}]$ and $\boldsymbol{u}_t = \text{col}[\boldsymbol{u}_t^{(1)}, \boldsymbol{u}_t^{(2)}]$, we can write it into one line

$$
\boldsymbol{x}_{t+1} = \boldsymbol{x}_t - \eta \cdot g_\mu(\boldsymbol{x}_t, \xi_t, \boldsymbol{u}_t) \cdot \boldsymbol{u}_t \quad\quad (24)
$$

Next, we define a new (virtual) zeroth-order gradient scalar without the compression step:

$$
\begin{aligned}
& g_\mu(\boldsymbol{x}_t, \xi_t, \boldsymbol{u}_t) \\
=& \frac{F\big(S_2(a_t^+; \boldsymbol{x}_t^{(2)} + \mu \boldsymbol{u}_t^{(2)})\big) - F\big(S_2(a_t^-; \boldsymbol{x}_t^{(2)} - \mu \boldsymbol{u}_t^{(2)})\big)}{\mu} \\
=& \frac{F\big(S_2(S_1(\xi_t; \boldsymbol{x}_t^{(1)} + \mu \boldsymbol{u}_t^{(1)}); \boldsymbol{x}_t^{(2)} + \mu \boldsymbol{u}_t^{(2)})\big) - F\big(S_2(S_1(\xi_t; \boldsymbol{x}_t^{(1)} - \mu \boldsymbol{u}_t^{(2)}); \boldsymbol{x}_t^{(2)} - \mu \boldsymbol{u}_t^{(2)})\big)}{\mu} \quad (25)
\end{aligned}
$$

It is not hard to see that $\mathbb{E}_{\xi_t, \boldsymbol{u}_t}[G_\mu(\boldsymbol{x}_t, \xi_t, \boldsymbol{u}_t)] = \nabla f_\mu(\boldsymbol{x}_t)$ using the Lemma 3 (a). Moreover, we define the difference between these two gradient scalars as:

$$
\Delta_\mu(\boldsymbol{x}_t, \xi_t, \boldsymbol{u}_t) := g_\mu(\boldsymbol{x}_t, \xi_t, \boldsymbol{u}_t) - \hat{g}_\mu(\boldsymbol{x}_t, \xi_t, \boldsymbol{u}_t) \quad\quad (26)
$$

Now, we arrive at

$$
\boldsymbol{x}_{t+1} = \boldsymbol{x}_t - \eta g_\mu(\boldsymbol{x}_t, \xi_t, \boldsymbol{u}_t) \cdot \boldsymbol{u}_t + \eta \Delta_\mu(\boldsymbol{x}_t, \xi_t, \boldsymbol{u}_t) \cdot \boldsymbol{u}_t \quad\quad (27)
$$

$$=\boldsymbol{x}_t - \eta G_\mu(\boldsymbol{x}_t, \xi_t, \boldsymbol{u}_t) + \eta \Delta_\mu(\boldsymbol{x}_t, \xi_t, \boldsymbol{u}_t) \cdot \boldsymbol{u}_t \tag{28}$$

When there is no ambiguities, we simply use $G_\mu$ and $\Delta_\mu$ to replace $G_\mu(\boldsymbol{x}_t, \xi_t, \boldsymbol{u}_t)$ and $\Delta_\mu(\boldsymbol{x}_t, \xi_t, \boldsymbol{u}_t)$ respectively.

*Proof of Theorem 1:*

We start with the Lipschitz condition:

$$\mathbb{E}\left[f_\mu(\boldsymbol{x}_{t+1})\right]$$

$$\leq f_\mu(\boldsymbol{x}_t) + \mathbb{E}\left[\langle \nabla f_\mu(\boldsymbol{x}_t), \boldsymbol{x}_{t+1} - \boldsymbol{x}_t \rangle\right] + \frac{L}{2}\mathbb{E}\left\|\boldsymbol{x}_{t+1} - \boldsymbol{x}_t\right\|^2$$

$$= f_\mu(\boldsymbol{x}_t) - \eta \langle \nabla f_\mu(\boldsymbol{x}_t), \mathbb{E}\left[G_\mu - \Delta_\mu \cdot \boldsymbol{u}_t\right]\rangle + \frac{L}{2}\eta^2 \mathbb{E}\left\|\hat{G}_\mu\right\|^2$$

$$\leq f_\mu(\boldsymbol{x}_t) - \eta \mathbb{E}\left[\langle \nabla f_\mu(\boldsymbol{x}_t), G_\mu \rangle\right] + \frac{\eta}{2\epsilon}\mathbb{E}\left\|\nabla f_\mu(\boldsymbol{x}_t)\right\|^2 + \frac{\eta\epsilon}{2}\left\|\mathbb{E}\left[\Delta_\mu \cdot \boldsymbol{u}_t\right]\right\|^2 + L\eta^2 \mathbb{E}\left\|\hat{G}_\mu\right\|^2 \tag{29}$$

$$= f_\mu(\boldsymbol{x}_t) - \eta \mathbb{E}\left[\langle \nabla f_\mu(\boldsymbol{x}_t), \nabla f_\mu(\boldsymbol{x}_t) - \nabla f_\mu(\boldsymbol{x}_t) + G_\mu \rangle\right] + \frac{\eta}{2\epsilon}\mathbb{E}\left\|\nabla f_\mu(\boldsymbol{x}_t)\right\|^2$$

$$+ \frac{\eta\epsilon}{2}\left\|\mathbb{E}\left[G_\mu - \hat{G}_\mu\right]\right\|^2 + L\eta^2 \mathbb{E}\left\|\hat{G}_\mu\right\|^2$$

$$= f_\mu(\boldsymbol{x}_t) - \eta \mathbb{E}\left\|\nabla f_\mu(\boldsymbol{x}_t)\right\|^2 - \eta \mathbb{E}\left[\langle \nabla f_\mu(\boldsymbol{x}_t), G_\mu - \nabla f_\mu(\boldsymbol{x}_t)\rangle\right] + \frac{\eta}{2\epsilon}\mathbb{E}\left\|\nabla f_\mu(\boldsymbol{x}_t)\right\|^2$$

$$+ \frac{\eta\epsilon}{2}\left\|\nabla f_\mu(\boldsymbol{x}_t) - \nabla \hat{f}_\mu(\boldsymbol{x}_t)\right\|^2 + L\eta^2 \mathbb{E}\left\|\hat{G}_\mu\right\|^2$$

$$= f_\mu(\boldsymbol{x}_t) + (\frac{\eta}{2\epsilon} - \eta)\mathbb{E}\left\|\nabla f_\mu(\boldsymbol{x}_t)\right\|^2 + \frac{\eta\epsilon}{2}\left\|\nabla f_\mu(\boldsymbol{x}_t) - \nabla \hat{f}_\mu(\boldsymbol{x}_t)\right\|^2 + L\eta^2 \mathbb{E}\left\|\hat{G}_\mu\right\|^2, \tag{30}$$

where (29) applies the Young's inequality on the cross term, where $\epsilon$ is an arbitrary positive number, and the Jensen's inequality on the last term.

Re-arranging (30), we get

$$\eta(1 - \frac{1}{2\epsilon})\mathbb{E}\left\|\nabla f_\mu(\boldsymbol{x}_t)\right\|^2 \leq f_\mu(\boldsymbol{x}_t) - \mathbb{E}\left[f_\mu(\boldsymbol{x}_{t+1})\right] + \frac{\eta\epsilon}{2}\left\|\nabla f_\mu(\boldsymbol{x}_t) - \nabla \hat{f}_\mu(\boldsymbol{x}_t)\right\|^2 + L\eta^2 \mathbb{E}\left\|\hat{G}_\mu\right\|^2. \tag{31}$$

Then, we sum up all (31) for all $t = 0, ..., T-1$ and establish

$$\eta(1 - \frac{1}{2\epsilon})\sum_{t=0}^{T-1}\mathbb{E}\left\|\nabla f_\mu(\boldsymbol{x}_t)\right\|^2$$

$$\leq f_\mu(\boldsymbol{x}_0) - f_\mu(\boldsymbol{x}_T) + \frac{\eta\epsilon}{2}\sum_{t=0}^{T-1}\left\|\nabla f_\mu(\boldsymbol{x}_t) - \nabla \hat{f}_\mu(\boldsymbol{x}_t)\right\|^2 + L\eta^2 \sum_{t=0}^{T-1}\mathbb{E}\left\|\hat{G}_\mu\right\|^2 \tag{32}$$

$$\leq f_\mu(\boldsymbol{x}_0) - f_\mu(\boldsymbol{x}^*) + \frac{\eta\epsilon}{2}\underbrace{\sum_{t=0}^{T-1}\left\|\nabla f_\mu(\boldsymbol{x}_t) - \nabla \hat{f}_\mu(\boldsymbol{x}_t)\right\|^2}_{A_1} + L\eta^2 \underbrace{\sum_{t=0}^{T-1}\mathbb{E}\left\|\hat{G}_\mu\right\|^2}_{A_2}, \tag{33}$$

where the last inequality follows from $f_\mu(\boldsymbol{x}^*) \leq f_\mu(\boldsymbol{x}_T)$.

Next, we process $A_1$ as follows. We apply (16) on the first term and the third term of (34). For the second term of (34), we obtain it by Lemma 4.

Now, we deal with $A_1$:

$$A_1 = \left\|\nabla f_\mu(\boldsymbol{x}_t) - \nabla \hat{f}_\mu(\boldsymbol{x}_t)\right\|^2$$

$$= \left\|\nabla f_\mu(\boldsymbol{x}_t) - \nabla f(\boldsymbol{x}_t) + \nabla f(\boldsymbol{x}_t) - \nabla \hat{f}(\boldsymbol{x}_t) + \nabla \hat{f}(\boldsymbol{x}_t) - \nabla \hat{f}_\mu(\boldsymbol{x}_t)\right\|^2$$

$$\leq 3 \left\| \nabla f_\mu(\boldsymbol{x}_t) - \nabla f(\boldsymbol{x}_t) \right\|^2 + 3 \left\| \nabla f(\boldsymbol{x}_t) - \nabla \hat{f}(\boldsymbol{x}_t) \right\|^2 + 3 \left\| \nabla \hat{f}(\boldsymbol{x}_t) - \nabla \hat{f}_\mu(\boldsymbol{x}_t) \right\|^2 \tag{34}$$

$$\leq \frac{3}{4} \mu^2 L^2 (d+3)^3 + 3(1 + C_{S_1}^2) L_2^2 \kappa^2 L_{S_1}^2 + \frac{3}{4} \mu^2 L^2 (d+3)^3 \tag{35}$$

$$= \frac{3}{2} \mu^2 L^2 (d+3)^3 + 3(1 + C_{S_1}^2) L_2^2 \kappa^2 L_{S_1}^2, \tag{36}$$

where we obtain (35) by Lemma 4.

Then, we start to deal with $A_2$.

$$A_2 = L\eta^2 \sum_{t=0}^{T-1} \mathbb{E} \left\| \hat{G}_\mu \right\|^2$$

$$\leq L\eta^2 \sum_{t=1}^{T} \left( \frac{\mu^2}{2} L^2 (d+6)^3 + 2(d+4) \mathbb{E} \left\| \hat{G} \right\|^2 \right)$$

$$\leq L\eta^2 \sum_{t=0}^{T-1} \left( \frac{\mu^2}{2} L^2 (d+6)^3 + 2(d+4) \left( \mathbb{E} \left\| \nabla \hat{f}(\boldsymbol{x}_t) \right\|^2 + \sigma^2 \right) \right)$$

$$= L\eta^2 \sum_{t=0}^{T-1} \left( \frac{\mu^2}{2} L^2 (d+6)^3 + 2(d+4) \left( \mathbb{E} \left\| \nabla \hat{f}(\boldsymbol{x}_t) - \nabla f(\boldsymbol{x}_t) + \nabla f(\boldsymbol{x}_t) \right\|^2 + \sigma^2 \right) \right)$$

$$\leq L\eta^2 \sum_{t=0}^{T-1} \left( \frac{\mu^2}{2} L^2 (d+6)^3 + 2(d+4) \left( 2\mathbb{E} \left\| \nabla \hat{f}(\boldsymbol{x}_t) - \nabla f(\boldsymbol{x}_t) \right\|^2 + 2\mathbb{E} \|\nabla f(\boldsymbol{x}_t)\|^2 + \sigma^2 \right) \right) \tag{37}$$

$$\leq L\eta^2 \sum_{t=0}^{T-1} \left( \frac{\mu^2}{2} L^2 (d+6)^3 + 2(d+4) \left( 2(1 + C_{S_1}^2) L_2^2 \kappa^2 L_{S_1}^2 + \sigma^2 \right) + 4(d+4) \mathbb{E} \|\nabla f(\boldsymbol{x}_t)\|^2 \right)$$

$$= L\eta^2 \sum_{t=0}^{T-1} \left( \frac{\mu^2}{2} L^2 (d+6)^3 + 2(d+4) \left( 2(1 + C_{S_1}^2) L_2^2 \kappa^2 L_{S_1}^2 + \sigma^2 \right) \right) + 4(d+4) L\eta^2 \sum_{t=0}^{T-1} \mathbb{E} \|\nabla f(\boldsymbol{x}_t)\|^2, \tag{38}$$

where we apply Lemma 4 on the term $\left\| \nabla \hat{f}(\boldsymbol{x}_t) - \nabla f(\boldsymbol{x}_t) \right\|^2$ of (37).

Putting all pieces above together, we establish:

$$(\eta - \frac{\eta}{2\epsilon}) \sum_{t=0}^{T-1} \mathbb{E} \|\nabla f_\mu(\boldsymbol{x}_t)\|^2 \leq f_\mu(\boldsymbol{x}_0) - f_\mu(\boldsymbol{x}^*) + \frac{\eta\epsilon}{2} \sum_{t=0}^{T-1} \left( \frac{3}{2} \mu^2 L^2 (d+3)^3 + 3(1 + C_{S_1}^2) L_2^2 \kappa^2 L_{S_1}^2 \right)$$

$$+ L\eta^2 \sum_{t=0}^{T-1} \left( \frac{\mu^2}{2} L^2 (d+6)^3 + 2(d+4) \left( 2(1 + C_{S_1}^2) L_2^2 \kappa^2 L_{S_1}^2 + \sigma^2 \right) \right)$$

$$+ 4(d+4) L\eta^2 \sum_{t=0}^{T-1} \mathbb{E} \|\nabla f(\boldsymbol{x}_t)\|^2 \tag{39}$$

By using (20) and re-arranging (39), we get

$$\left( \frac{\eta}{2} (1 - \frac{1}{2\epsilon}) - 4(d+4) L\eta^2 \right) \sum_{t=0}^{T-1} \mathbb{E} \|\nabla f(\boldsymbol{x}_t)\|^2$$

$$\leq f(\boldsymbol{x}_0) - f(\boldsymbol{x}^*) + \mu^2 L d + \frac{\eta\epsilon}{2} \sum_{t=0}^{T-1} \left( \frac{3}{2} \mu^2 L^2 (d+3)^3 + 3(1 + C_{S_1}^2) L_2^2 \kappa^2 L_{S_1}^2 \right)$$

$$+ L\eta^2 \sum_{t=0}^{T-1} \left( \frac{\mu^2}{2} L^2 (d+6)^3 + 2(d+4) \left( 2(1 + C_{S_1}^2) L_2^2 \kappa^2 L_{S_1}^2 + \sigma^2 \right) \right)$$

$$+ \eta(1 - \frac{1}{2\epsilon}) \sum_{t=0}^{T-1} \frac{\mu^2}{4} L^2(d+3)^3 \tag{40}$$

We select $\epsilon = 1$ and further simplify

$$\left(\eta - 16(d+4)L\eta^2\right) \sum_{t=0}^{T-1} \mathbb{E}\|\nabla f(\boldsymbol{x}_t)\|^2 \leq 4\big(f(\boldsymbol{x}_0) - f(\boldsymbol{x}^*)\big) + 4\mu^2 Ld + \eta\big(3\mu^2 L^2(d+3)^3$$
$$+ L\eta^2 T \left(2\mu^2 L^2(d+6)^3 + 8(d+4)\Big(2(1+C_{S_1}^2)L_2^2\kappa^2 L_{S_1}^2 + \sigma^2\Big)\right)$$
$$+ 6(1 + C_{S_1}^2)L_2^2\kappa^2 L_{S_1}^2)T + \frac{1}{2}\eta\mu^2 L^2(d+3)^3 T \tag{41}$$

We assume that $\frac{1}{2}\eta \leq \big(\eta - 16(d+4)L\eta^2\big)$ and then get $\eta \leq \frac{1}{32(d+4)L}$. Then, we can further simplify the inequality above.

$$\frac{1}{T} \sum_{t=0}^{T-1} \mathbb{E}\|\nabla f(\boldsymbol{x}_t)\|^2 \leq \frac{8\big(f(\boldsymbol{x}_0) - f(\boldsymbol{x}^*)\big)}{T\eta} + \frac{8\mu^2 Ln}{T\eta} + 7\mu^2 L^2(d+3)^3 + 12(1+C_{S_1}^2)L_2^2\kappa^2 L_{S_1}^2$$
$$+ 4L\eta\left(\mu^2 L^2(d+6)^3 + 4(d+4)\Big(2(1+C_{S_1}^2)L_2^2\kappa^2 L_{S_1}^2 + \sigma^2\Big)\right) \tag{42}$$

Assuming that $\eta = \mathcal{O}(\frac{1}{\sqrt{Td}})$, then we obtain

$$\frac{1}{T} \sum_{t=0}^{T-1} \mathbb{E}\|\nabla f(\boldsymbol{x}_t)\|^2 = \mathcal{O}\left(\frac{D\sqrt{d}}{\sqrt{T}}\right) + \mathcal{O}\left(\frac{\mu^2 d^{\frac{3}{2}}}{\sqrt{T}}\right) + \mathcal{O}\Big(\mu^2(d+3)^3\Big) + \mathcal{O}\Big((1+C_{S_1}^2)L_2^2\kappa^2 L_{S_1}^2\Big)$$
$$+ \mathcal{O}\left(\frac{\mu^2(d+6)^3}{\sqrt{Td}}\right) + \mathcal{O}\left(\frac{(d+4)(1+C_{S_1}^2)L_2^2\kappa^2 L_{S_1}^2}{\sqrt{Td}}\right) + \mathcal{O}\left(\frac{(d+4)\sigma^2}{\sqrt{Td}}\right), \tag{43}$$

where $D = f(\boldsymbol{x}_0) - f(\boldsymbol{x}^*)$.

Further, we set $\mu \leq \frac{1}{(d+6)\sqrt{T}}$, then we can get

$$\frac{1}{T} \sum_{t=0}^{T-1} \mathbb{E}\|\nabla f(\boldsymbol{x}_t)\|^2 = \mathcal{O}\left(\frac{D\sqrt{d}}{\sqrt{T}}\right) + \mathcal{O}\left(\frac{1}{T\sqrt{d}}\right) + \mathcal{O}\left(\frac{d}{T}\right) + \mathcal{O}\Big((1+C_{S_1}^2)L_2^2\kappa^2 L_{S_1}^2\Big)$$
$$+ \mathcal{O}\left(\frac{\sqrt{d}}{T^{\frac{3}{2}}}\right) + \mathcal{O}\left(\frac{\sqrt{d}}{\sqrt{T}}(1+C_{S_1}^2)L_2^2\kappa^2 L_{S_1}^2\right) + \mathcal{O}\left(\frac{\sqrt{d}}{\sqrt{T}}\sigma^2\right)$$

Further, we simplify it, then we can get

$$\frac{1}{T} \sum_{t=0}^{T-1} \mathbb{E}\|\nabla f(\boldsymbol{x}_t)\|^2 = \mathcal{O}\left(\frac{D\sqrt{d}}{\sqrt{T}}\right) + \mathcal{O}\left(\frac{d}{T}\right) + \mathcal{O}\left((1+C_{S_1}^2)L_2^2\kappa^2 L_{S_1}^2\left(\frac{\sqrt{d}}{\sqrt{T}}+1\right)\right) + \mathcal{O}\left(\frac{\sqrt{d}}{\sqrt{T}}\sigma^2\right)$$

$\blacksquare$

## C.3 Proof of Theorem 2

**Assumption 6 (Lipschitz Gradients)** *For multi-node cases ($M > 2$), we suppose that*

- *$f(\cdot)$ has $L$-Lipschitz gradient,*

- *$f \circ S_M \circ \cdots S_{i+1}$ has a $L_{f \circ S_M \circ \cdots S_{i+1}}$-Lipschitz gradient, and its gradient is bounded by $C_{f \circ S_M \circ \cdots S_{i+1}}$ for all $i = 1, ..., M-1$,*

- *The submodel $S_i$ is $L_{S_i}$-Lipschitz, and its gradient is bounded by $C_{S_i}$, for all $i = 1, ..., M$.*

**Lemma 5 (Distance between Gradients of Uncompressed and Compressed Activations, M>2)**
*For Alg. 3, the difference between gradients of uncompressed and compressed activations can be bounded by a constant term as follows:*

$$\left\|\nabla f(\boldsymbol{x}_t) - \nabla \hat{f}(\boldsymbol{x}_t)\right\|^2 \leq 2M\kappa^2 L_S^2 \Psi$$

*Proof of lemma 5:*

For simplicity in the following proof, we denote that

$$\bar{S}_i = S_i(\cdots(S_2(S_1(\xi, \boldsymbol{x}_t^{(1)}); \boldsymbol{x}_t^{(2)}); \cdots); \boldsymbol{x}_t^{(i)}) \text{ and } \bar{m}_i = S_i(\cdots(\mathcal{C}(S_1(\xi; \boldsymbol{x}_t^{(1)})); \cdots); \boldsymbol{x}_t^{(i)}).$$

Then, we bound the gradient difference as follows:

$$\left\|\nabla f(\boldsymbol{x}_t) - \nabla \hat{f}(\boldsymbol{x}_t)\right\|^2$$

$$= \left\|\nabla_{\boldsymbol{x}^{(1)}} f(\boldsymbol{x}_t) - \nabla_{\boldsymbol{x}^{(1)}} \hat{f}(\boldsymbol{x}_t)\right\|^2 + \cdots + \left\|\nabla_{\boldsymbol{x}^{(M)}} f(\boldsymbol{x}_t) - \nabla_{\boldsymbol{x}^{(M)}} \hat{f}(\boldsymbol{x}_t)\right\|^2$$

$$= \sum_{i=1}^{M} \left\|\nabla_{S_i} (F \circ S_M \circ \cdots \circ S_{i+1})\big|_{(\bar{S}_i, \cdots, \boldsymbol{x}_t^{(M)})} \cdot \nabla_{\boldsymbol{x}^{(2)}} S_i\big|_{(\bar{S}_{i-1}, \boldsymbol{x}_t^{(i)})}\right.$$

$$\left. - \nabla_{S_i} (F \circ S_M \circ \cdots \circ S_{i+1})\big|_{(\bar{m}_i, \boldsymbol{x}_t^{(i+1)}, \cdots, \boldsymbol{x}_t^{(M)})} \cdot \nabla_{\boldsymbol{x}^{(i)}} S_i\big|_{(\bar{m}_{i-1}, \boldsymbol{x}_t^{(i)})}\right\|^2$$

$$\leq (1 + 2C_{S_{M-1}}^2) L_{f \circ S_M}^2 \kappa^2 L_{S_{M-1}}^2 + 2\kappa^2 \sum_{i=1}^{M-2} (C_{S_{M-2}}^2 L_{f \circ S_M \circ \cdots \circ S_{i+1}}^2 + C_{f \circ S_M \circ \cdots \circ S_{i+2}}^2 L_{S_{i+1}}^2) L_{S_i}^2$$

$$\leq 2M\kappa^2 L_S^2 \underbrace{\max\left\{(1 + 2C_{S_{M-1}}^2) L_{f \circ S_M}^2 \kappa^2, \max_{i \in [M-2]} \{C_{S_{M-2}}^2 L_{f \circ S_M \circ \cdots \circ S_{i+1}}^2 + C_{f \circ S_M \circ \cdots \circ S_{i+2}}^2 L_{S_{i+1}}^2\}\right\}}_{\Psi},$$

where $L_S^2 = \max\{L_{S_1}^2, \cdots, L_{S_M}^2\}$. ∎

**Theorem 2 (Convergence of `SparQ` under Non-Convexity, $M>2$)** *For Alg. 3, under assumptions 1, 2, 3, 5 and 6, if the number of computing nodes $M > 2$ and learning rate $\eta \leq 1/32(d+4)L$, then the sequence $\{\boldsymbol{x}_t\}$ generated by `SparQ` satisfies*

$$\frac{1}{T} \sum_{t=0}^{T-1} \mathbb{E}\|\nabla f(\boldsymbol{x}_t)\|^2 \leq \frac{8(f(\boldsymbol{x}_0) - f(\boldsymbol{x}^*))}{T\eta} + \frac{8\mu^2 Ln}{T\eta} + 7\mu^2 L^2 (d+3)^3 + 12M\kappa^2 L_S^2 \Psi$$

$$+ 4L\eta \left(\mu^2 L^2 (d+6)^3 + 4(d+4)(4M\kappa^2 L_S^2 \Psi + \sigma^2)\right),$$

*where the definitions of $L_S$ and $\Psi$ can be found in the following proof.*

*Proof of Theorem 2:*

Here, we only provide a rough proof because we use the same proof framework as the proof of Theorem 1. The key differences are using Lemma 5 in inequalities (34) and (37).

Hence, $A_1$ and $A_2$ will be modified as follows:

$$A_1 = \left\|\nabla f_\mu(\boldsymbol{x}_t) - \nabla \hat{f}_\mu(\boldsymbol{x}_t)\right\|^2$$

$$= \left\|\nabla f_\mu(\boldsymbol{x}_t) - \nabla f(\boldsymbol{x}_t) + \nabla f(\boldsymbol{x}_t) - \nabla \hat{f}(\boldsymbol{x}_t) + \nabla \hat{f}(\boldsymbol{x}_t) - \nabla \hat{f}_\mu(\boldsymbol{x}_t)\right\|^2$$

$$\leq 3 \|\nabla f_\mu(\boldsymbol{x}_t) - \nabla f(\boldsymbol{x}_t)\|^2 + 3 \left\|\nabla f(\boldsymbol{x}_t) - \nabla \hat{f}(\boldsymbol{x}_t)\right\|^2 + 3 \left\|\nabla \hat{f}(\boldsymbol{x}_t) - \nabla \hat{f}_\mu(\boldsymbol{x}_t)\right\|^2$$

$$\leq \frac{3}{4}\mu^2 L^2(d+3)^3 + 6M\kappa^2 L_S^2\Psi + \frac{3}{4}\mu^2 L^2(d+3)^3 \tag{44}$$

$$= \frac{3}{2}\mu^2 L^2(d+3)^3 + 6M\kappa^2 L_S^2\Psi,$$

where we obtain (44) by Lemma 5. Then, we start to address $A_2$:

$$A_2 = L\eta^2 \sum_{t=0}^{T-1} \mathbb{E} \left\| \hat{G}_\mu \right\|^2$$

$$\leq L\eta^2 \sum_{t=1}^{T} \left( \frac{\mu^2}{2}L^2(d+6)^3 + 2(d+4)\mathbb{E} \left\| \hat{G} \right\|^2 \right)$$

$$\leq L\eta^2 \sum_{t=0}^{T-1} \left( \frac{\mu^2}{2}L^2(d+6)^3 + 2(d+4) \left( \mathbb{E} \left\| \nabla \hat{f}(\boldsymbol{x}_t) \right\|^2 + \sigma^2 \right) \right)$$

$$= L\eta^2 \sum_{t=0}^{T-1} \left( \frac{\mu^2}{2}L^2(d+6)^3 + 2(d+4) \left( \mathbb{E} \left\| \nabla \hat{f}(\boldsymbol{x}_t) - \nabla f(\boldsymbol{x}_t) + \nabla f(\boldsymbol{x}_t) \right\|^2 + \sigma^2 \right) \right)$$

$$\leq L\eta^2 \sum_{t=0}^{T-1} \left( \frac{\mu^2}{2}L^2(d+6)^3 + 2(d+4) \left( 2\mathbb{E} \left\| \nabla \hat{f}(\boldsymbol{x}_t) - \nabla f(\boldsymbol{x}_t) \right\|^2 + 2\mathbb{E} \left\| \nabla f(\boldsymbol{x}_t) \right\|^2 + \sigma^2 \right) \right)$$

$$\leq L\eta^2 \sum_{t=0}^{T-1} \left( \frac{\mu^2}{2}L^2(d+6)^3 + 2(d+4) \left( 4M\kappa^2 L_S^2\Psi + \sigma^2 \right) + 4(d+4)\mathbb{E} \left\| \nabla f(\boldsymbol{x}_t) \right\|^2 \right)$$

$$= L\eta^2 \sum_{t=0}^{T-1} \left( \frac{\mu^2}{2}L^2(d+6)^3 + 2(d+4)(4M\kappa^2 L_S^2\Psi + \sigma^2) \right) + 4(d+4)L\eta^2 \sum_{t=0}^{T-1} \mathbb{E} \left\| \nabla f(\boldsymbol{x}_t) \right\|^2,$$

Then, the subsequent steps are the same as the proof of Theorem 1, so we skip them and directly arrive at

$$\left( \eta - 16(d+4)L\eta^2 \right) \sum_{t=0}^{T-1} \mathbb{E} \left\| \nabla f(\boldsymbol{x}_t) \right\|^2 \leq 4 \left( f(\boldsymbol{x}_0) - f(\boldsymbol{x}^*) \right) + 4\mu^2 Ld + \eta \left( 3\mu^2 L^2(d+3)^3 \right.$$

$$+ L\eta^2 T \left( 2\mu^2 L^2(d+6)^3 + 8(d+4)(4M\kappa^2 L_S^2\Psi + \sigma^2) \right) + 6M\kappa^2\Psi)T + \frac{1}{2}\eta\mu^2 L^2(d+3)^3 T$$

We assume that $\frac{1}{2}\eta \leq \left( \eta - 16(d+4)L\eta^2 \right)$ and then get $\eta \leq \frac{1}{32(d+4)L}$. Then, we can further simplify the inequality above and obtain

$$\frac{1}{T} \sum_{t=0}^{T-1} \mathbb{E} \left\| \nabla f(\boldsymbol{x}_t) \right\|^2 \leq \frac{8 \left( f(\boldsymbol{x}_0) - f(\boldsymbol{x}^*) \right)}{T\eta} + \frac{8\mu^2 Ln}{T\eta} + 7\mu^2 L^2(d+3)^3 + 12M\kappa^2 L_S^2\Psi$$

$$+ 4L\eta \left( \mu^2 L^2(d+6)^3 + 4(d+4)(4M\kappa^2 L_S^2\Psi + \sigma^2) \right)$$

■

