# OpenReview forum: "Efficient Fine-Tuning of Large Language Models with Zeroth-Order Model Parallelism"
_TMLR — Under review for TMLR_

### Review · Reviewer_uZ9a · 2026-05-21

**Summary Of Contributions:**

This paper proposes SparQ, a framework combining zeroth-order (ZO) optimization with model parallelism (MP) for LLM fine-tuning. The motivation is that first-order MP suffers from high communication (gradient/activation exchange) and memory (cached states) costs. SparQ addresses both via three components: (1) ZO-SGD eliminates gradient storage and transmission; (2) 4-bit quantization applied after activation functions induces high sparsity (1–30% nonzero), even for smooth activations like GELU/SwiGLU that are otherwise dense; (3) the model is split immediately after an activation function so that transmitted activations are sparse and can be encoded as (value, index) pairs. The authors prove a sublinear non-convex convergence rate matching centralized ZO-SGD, and report up to 3x memory and 50–70% communication reduction over FO-SGD, AQ-SGD, and MeZO on SST-2/WIC/RTE across OPT, GPT-2, and Llama-1B.

### Key Strengths

- Combining ZO with MP is a natural idea that has not been systematically explored, and the deliberate placement of the split at a sparsity-rich layer is a genuine insight.
- The empirical finding that 4-bit quantization drives GELU/SwiGLU activations toward high sparsity is interesting independently of the framework.
- Convergence analysis is provided for both two-node and multi-node settings under standard assumptions.
- Ablations on split location and quantization bit-width are the right questions to ask.

### Key Weaknesses

- The need for MP is not defended at the evaluated scale: by the paper's own numbers, MeZO fits OPT-6.7B on a single modern GPU, so it is unclear when SparQ is necessary rather than merely convenient.
- The "comparable performance" framing understates the results. On RTE and WIC, SparQ trails FO-SGD by 6–13 points across multiple models, with several numbers near majority-class baselines.
- Communication savings are reported per iteration, not per unit of progress. Since ZO typically needs many more steps than FO to reach a target accuracy, total communication to a target is the appropriate metric and is not provided.
- AQ-SGD appears to underperform its original report (e.g., on RTE) without explanation or tuning details, making the head-to-head comparison hard to trust.
- A trivial baseline—magnitude-based activation sparsification without quantization—is missing, so the specific role of quantization is not cleanly isolated.
- The largest model is OPT-6.7B, which is small relative to the regime where MP is genuinely required; the "large language models" framing is not backed by experiments at that scale.
- Algorithm 1 uses a shared random seed across nodes, while the analysis writes the perturbation vectors as separately drawn on each node. This inconsistency should be reconciled.
- Figure 1 uses only SST-2, the benchmark on which all methods are closest, making the Pareto picture look more favorable than Table 2 supports.

**Audience:**

Yes

**Audience Explanation:**

Several findings in this paper would be of interest to subsets of the TMLR audience:
- Researchers working on ZO optimization for LLMs would find the integration of ZO with model parallelism a natural and previously unexplored direction. Even if the practical case for this combination is not fully made, the formulation itself is a useful reference point.
- The activation compression and efficient training community would likely find the empirical observation in Figure 3 useful on its own: that 4-bit quantization drives GELU and SwiGLU activations—normally nearly dense—toward high sparsity. This is a non-obvious finding with potential implications beyond the specific framework proposed here.
- Researchers studying split learning and pipeline parallelism may be interested in the principle of deliberately placing the partition at a sparsity-rich layer, which is a transferable design idea independent of whether ZO is used.
- The convergence proof for ZO with biased activation compression in a model-parallel setting is a modest but reasonable theoretical contribution that may be referenced by future work in this space.

**Broader Impact Concerns:**

None. The paper focuses on efficiency improvements (memory and communication) for fine-tuning existing language models and does not introduce new capabilities, data, or deployment scenarios that raise distinct ethical concerns beyond those already present in the LLM fine-tuning literature. The existing Broader Impact Statement, which notes no specific negative societal consequences, is adequate for the scope of the work.

**Claims And Evidence:**

No

**Claims Explanation:**

Partially. Several central claims are not adequately supported by the evidence presented.
### Claims that are reasonably supported:

- The empirical observation that 4-bit quantization induces high activation sparsity across ReLU, GELU, and SwiGLU is clearly demonstrated in Figure 3 and is convincing.
- The convergence analysis follows standard ZO + compression proof techniques, and the assumptions used are reasonable for this literature.
- The memory reduction claim (around 3x over FO-SGD) is plausible and consistent with what MeZO already established for ZO methods generally—most of this saving comes from ZO itself rather than from SparQ's specific contributions.

### Claims that are not adequately supported:

- "Comparable performance" to baselines. This is the most problematic claim. On RTE and WIC, SparQ trails FO-SGD by 6–13 accuracy points across multiple models (e.g., OPT-1.3B/RTE: 57.1% vs. 70.8%; GPT-2/RTE: 52.3% vs. 63.1%), with several results near majority-class baselines. The text repeatedly characterizes these gaps as "comparable," which does not match the numbers in Table 2. The claim holds only on SST-2, which is the easiest benchmark and the one Figure 1 selectively uses.
- "50–70% communication reduction." Reported per iteration, not per unit of progress. Since ZO methods typically require substantially more steps than FO to reach a target loss, the relevant quantity is total communication to a target accuracy, which is not provided. Without this, the headline number is not directly comparable to FO baselines.
- Superiority over AQ-SGD. AQ-SGD underperforms its original paper's reported accuracy by a large margin on several tasks (e.g., RTE), with no tuning details or explanation. This makes the comparison difficult to trust.
- Causal role of quantization in the sparsity story. The framework is named for "quantization-induced sparsity," but no baseline isolates quantization from generic magnitude-based activation sparsification. The specific contribution of quantization, as opposed to any thresholding scheme, is therefore not demonstrated.
- Applicability to "large language models." The largest model evaluated is OPT-6.7B, which fits on a single modern GPU under ZO. The paper does not show results at scales where MP is genuinely required, so the title-level claim is not supported by the experiments.
Theory–algorithm consistency. Algorithm 1 uses a shared random seed across nodes for parameter perturbation, while the analysis writes the perturbation vectors as separately drawn on each node. The paper does not reconcile this, leaving uncertainty about which object the convergence theorem actually applies to.

**Requested Changes:**

- Broaden Figure 1 beyond SST-2. Currently it uses only the easiest benchmark, on which all methods are closest in accuracy. Including RTE or WIC would give a more honest Pareto picture and would help readers calibrate the trade-offs.
- Report variance across seeds. Several accuracy numbers in Table 2 sit close to majority-class baselines (e.g., WIC at around 53%). Without multi-seed standard deviations, it is unclear which differences are meaningful.
- Discuss the bias term in Lemma 1 empirically. Remark 2 notes that the compression-induced term is a constant independent of T, meaning the algorithm converges to a biased solution. An empirical study of how this bias scales with quantization level and split position would strengthen the link between theory and practice.
- Clarify the relationship to Sparse MeZO and similar ZO+sparsity methods. The related work mentions these briefly but does not discuss why the activation-sparsity approach here is fundamentally different from parameter-sparsity approaches, beyond noting that the latter is not MP-compatible.
- Improve writing in several places. Section 3.1 and parts of the introduction repeat the same points multiple times. Tightening the prose would help, especially given that the contribution is conceptually simple once the three pieces are stated.

---

> ### Author Response · Authors · 2026-06-25
>
> We sincerely appreciate the reviewer for the constructive comments, and we would like to reply to that as follows.
>
> > About the necessity of MP at the evaluated scale.
>
> **Our response:** Thank you so much for your comment. We will clarify that our motivation does not assume that every user has access to a high-memory modern GPU. Although MeZO can fit OPT-6.7B on one sufficiently large GPU in our evaluated setting, many practical users may instead have access to multiple smaller GPUs, each of which lacks enough memory to hold the full model and training-time states. In such cases, even if the aggregate memory across devices is sufficient, model parallelism becomes necessary rather than merely convenient.
>
> SparQ is designed for this practical setting. When the model must be partitioned across devices, activation communication becomes a key bottleneck, especially for model-parallel fine-tuning. SparQ addresses this by combining zeroth-order fine-tuning with quantized activation communication and sparsity-aware split placement, thereby enabling memory-constrained users to fine-tune models that would not fit on any single available device.
>
> > The "comparable performance" framing understates the results...
>
> **Our response:** Thank you for raising this point. We will revise the wording to avoid implying that SparQ is directly comparable to FO-SGD in accuracy. FO-SGD uses first-order gradients and serves in our experiments as a strong upper-reference, rather than as the primary baseline that SparQ is designed to match. The main target setting of SparQ is communication- and memory-efficient model-parallel fine-tuning. From this perspective, the most relevant baseline is AQ-SGD since it also considers compressed communication in a model-parallel training setting. MeZO/ZO-SGD provides an extra ZO performance reference, but it does not address the activation communication bottleneck under model parallelism. In the revision, whenever we refer to comparable performance, we will explicitly identify the corresponding baseline so that the intended comparison is clear and does not overstate our results.
>
> > About reported communication cost.
>
> **Our response:** Thank you for your comment. We believe that there is a misunderstanding regarding how communication cost is reported in our experiments. The communication costs reported in our experiment results are not per-iteration costs. Instead, they are the total cumulative communication costs over the full fine-tuning process, measured up to the final reported test accuracy. Therefore, these numbers already account for the number of optimization steps needed by each method to reach its reported final performance.
>
> In the revised manuscript, we will explicitly state that the reported communication cost is the total cumulative communication cost over the complete fine-tuning run, rather than the cost of a single iteration. We will also distinguish the per-iteration communication analysis from the end-to-end experimental communication measurements to avoid ambiguity.
>
> > AQ-SGD appears to underperform its original report (e.g., on RTE) ... | Superiority over AQ-SGD...
>
> **Our response:** Thank you for raising this concern. There appears to be a misunderstanding regarding the comparison to the original AQ-SGD report. To our knowledge, the original AQ-SGD paper does not report results on RTE. Its LLM fine-tuning experiments evaluate DeBERTa-1.5B on QNLI and CoLA, and GPT2-1.5B on WikiText2 and arXiv abstracts. Therefore, our AQ-SGD results on RTE are not underperforming a previously reported AQ-SGD RTE result since such a result is not provided in the original paper.
>
> Our AQ-SGD results are obtained by re-evaluating AQ-SGD under our unified experimental protocol, using the same datasets, model-parallel setting, batch size, and evaluation procedure as the other baselines. In Section 5.1, we describe our experiment setup for AQ-SGD: use 4-bit quantization to compress activations for AQ-SGD and 8-bit quantization to compress backward gradients for AQ-SGD.

---

> ### Author Response · Authors · 2026-06-25
>
> > A trivial baseline - magnitude-based activation sparsification without quantization... | Causal role of quantization in the sparsity story...
>
> **Our response:** Thank you for the comment. We would like to clarify that the role of quantization in SparQ is not equivalent to generic magnitude-based sparsification. Magnitude-based activation sparsification only reduces the number of transmitted nonzero entries: small-magnitude values are set to zero, but the retained nonzero values are still transmitted in high precision, e.g., FP16/FP32. In contrast, quantization in SparQ has two effects: it induces sparsity by mapping small activation values to zero, and it also compresses the remaining nonzero values into low-bit representations. Therefore, SparQ reduces communication through both fewer nonzero entries and fewer bits per retained value.
>
> This distinction is important. If magnitude-based sparsification uses the same nonzero ratio as SparQ, its communication cost can still be higher because each retained value requires more bits. If it tries to match SparQ’s communication budget, it must drop more activation entries, which can hurt accuracy. Moreover, choosing a robust magnitude threshold is nontrivial because activation distributions vary across layers, models, tasks, and training stages. This often requires layer-wise or dynamic threshold tuning. In comparison, the quantization level, such as 4-bit, is a simpler and more standard design choice.
> We also provide empirical evidence that split placement is critical. Placing the split between Transformer blocks leads to much higher communication cost and no accuracy benefit compared with SparQ's split-after-activation strategy.
>
> Table 1: Test Accuracy and Communication Cost Comparison. For "Split After Activation Functions(SparQ)" experiments, we use 4-bit quantization. For "Split Between Blocks" experiments, we use less aggressive magnitude-based sparsification.
> |Model|Dataset|Split Between Blocks|Split After Activation Functions(SparQ)|
> |-|-|-|-|
> |OPT-125M|SST-2|83.26% (4.1 GB)|84.58% (0.5 GB)|
> |OPT-125M|WIC|51.72% (4.7 GB)|53.42% (1.2 GB)|
> |OPT-125M|RTE|52.70% (5.3 GB)|53.27% (2.6 GB)|
> |OPT-1.3B|SST-2|90.51% (11.4 GB)|92.34% (1.2 GB)|
> |OPT-1.3B|WIC|55.60% (11.8 GB)|55.62% (3.1 GB)|
> |OPT-1.3B|RTE|56.80% (12.1 GB)|57.13% (7.0 GB)|
>
> These results show that SparQ’s benefit comes from the combination of sparsity-aware split placement and quantization-induced compression, rather than from arbitrary activation sparsification alone.
>
> > Algorithm 1 uses a shared random seed across nodes...
>
> **Our response:** Thank you for your comment. The intended interpretation is that $u_t^{(1)}$ and $u_t^{(2)}$ in the analysis are not independently sampled perturbation vectors. Rather, they are the local blocks of the same global perturbation vector, $u_t = \mathrm{col}[u_t^{(1)}, u_t^{(2)}]$, corresponding to the two model partitions. Algorithm 1 uses a shared random seed to let different nodes reconstruct their corresponding blocks of this same global perturbation direction without explicitly communicating the full vector. In the revised manuscript, we will make this notation explicit in both Algorithm 1 and the theoretical analysis.
>
> > About Figure 1.
>
> **Our response:** Thank you for the comment. Figure 1 is intended as a representative visualization of the memory-communication-accuracy trade-off on SST-2, rather than a summary of all benchmark tasks. We selected SST-2 for the main figure to provide a compact comparison across the three model families used in our experiments. The complete task-level results, including WIC and RTE, are reported in Table 2. To avoid any impression that the Pareto visualization is based only on the most favorable benchmark, we will also add corresponding Pareto plots for WIC in our updated Figure 1 in our updated paper.
>
> > Report variance across seeds...
>
> **Our response:** Thank you for the suggestion. Actually, in our original paper, our results were ontained given multiple seeds. Hence, in the revised manuscript, we report multi-seed results for all experiment results to make the comparison statistically clearer.

---

> ### Author Response · Authors · 2026-06-25
>
> > Discuss the bias term in Lemma 1 empirically. Remark 2 notes that the compression-induced term...
>
> **Our response:** Thank you for the comment. The compression-induced term in Lemma 1 captures the discrepancy between the gradients of the uncompressed and compressed objectives. This term depends on the compressor error \(\kappa\), and therefore reflects the possible bias introduced by activation compression.
> We note that this effect is already empirically reflected in our existing ablations. Table 1 studies different quantization levels from 1-bit to 8-bit. More aggressive quantization reduces communication but can slightly affect accuracy, while 4-bit provides a favorable balance. This directly corresponds to changing the compression strength, and hence the practical size of the compression-induced bias.
> We also study split placement. Fig. 4 and the split-position results show that placing the split after activation functions preserves trainability and accuracy, while splitting between blocks leads to worse compressed representations and higher communication cost. This is consistent with Lemma 1: the impact of compression depends on the activation representation at the split point.
> Finally, SparQ closely matches MeZO/ZO-SGD across models and tasks, indicating that the compression-induced bias is small in practice under our chosen 4-bit split-after-activation design.
>
> > Clarify the relationship to Sparse MeZO and similar ZO+sparsity methods.
>
> **Our response:** Thank you for the suggestion. We have expanded the related work discussion to clarify the distinction between SparQ and Sparse MeZO-/SensZOQ-style ZO+sparsity methods. The key difference is the object and purpose of sparsity. Sparse MeZO and related methods exploit sparsity in the parameter space, for example by perturbing or updating only a subset of model parameters. SensZOQ similarly focuses on static sensitive-parameter selection and weight quantization to enable memory-efficient ZO fine-tuning. Their goal is mainly to reduce the effective optimization dimension, computation, or memory cost of ZO fine-tuning. In contrast, SparQ exploits sparsity in the activation space. The sparse objects in SparQ are the intermediate activations transmitted across model partitions, and the goal is to reduce the communication cost introduced by model parallelism.
>
> This distinction is important because parameter sparsity and weight quantization do not directly address the activation communication bottleneck in model-parallel training. Even if only a sparse subset of parameters is perturbed or updated, or the model weights are quantized for memory savings, the intermediate activations at split layers still need to be communicated between devices. SparQ specifically targets this cross-partition communication by inducing and exploiting activation sparsity through quantization and sparsity-aware split placement.
>
> > Improve writing in several places...
>
> **Our response:** Thank you for the suggestion. Section 3.1 was included because the connection between activation sparsity and communication reduction is central to SparQ and is important for understanding why the split placement and sparse encoding are effective. We therefore keep this section, but make it more concise.

---

### Review · Reviewer_QsLp · 2026-06-01

**Summary Of Contributions:**

This paper advances the state of the art in resource-constrained LLM fine-tuning by configuring a zeroth-order (ZO) optimization recipe designed to use a minimal amount of memory per device and minimal communication bandwidth during between devices. Specifically, the paper introduces a recipe for the pipeline-parallel (PP) form of model parallelism (MP) that leverages quantization and the natural sparsity of quantized activation patterns of pretrained LLMs to dramatically compress the data being transmitted between devices.

The key contributions are a set of careful design decisions around the implementation of model-parallel (MP) distributed ZO fine-tuning that come together in an empirically fast and efficient implementation:
- using pipeline-parallel model parallelism (gives simple and sparsity-splittable communication patterns)
- using ZO-SGD optimization (ZO avoids storing activations and compresses backwards pass communication to scalar values only, SGD avoids requiring the storage of optimizer state in memory)
- using activation quantization to compress model activations passed between devices and induce sparsity in nearly-zero activation patterns produced by SwiGLU/GELU activation functions
- using sparse encoding to further compress quantized model activations passed between devices
- analyzing the sparsity patterns of quantized model activations to determine PP model splits around points of high sparsity that induce more compressible activations

**Additional Comments:**

I would like to apologize for providing a delayed review. I had lost track of my review obligation in a mistake that was entirely my own fault.

I would also like to note that I did not review Appendix C.

**Audience:**

Yes

**Audience Explanation:**

Efficient LLM fine-tuning seems pretty clearly to be a topic of significant interest to the TMLR community, and I believe the results of this paper represent a compelling advance in the state of the art.

**Claims And Evidence:**

Yes

**Claims Explanation:**

The authors give a clear exposition of the theoretical advantages of each aspect of their chosen design. The authors provide a theoretical analysis demonstrating that the approximations introduced by their design do not alter the asymptotic convergence rate of the theoretical convergence analysis ZO-SGD. The authors give solid experimental evidence of the practical impact of their methodology as well.

**Requested Changes:**

I was very impressed at the expository clarity of this paper. I believe this comes from the choice to, (beyond the "my method is better/faster/cheaper" headline result as Figure 1 that has become standard in ML publication), keep the Introduction short and to-the-point (rather than becoming an extensive "sales pitch") and instead spend more time in the Formulations and Preliminaries section building up reader understanding of the thinking behind the SparQ method. I would instead like to use this section to request that the authors make efforts to maintain this effective structure through any edits they may make before final publication.

If I were to make a couple small requests, one would be to make Figure 4 less confusing. It is not clear from the caption of text content where the activation pattern comes from or what aspects of the visualization are supposed to explain why some configurations are trainable versus not trainable. The text gives some hints that the pre-quantization sparsity of ReLU and the increased number of nonzero elements post-quantization for SwiGLU and GELU may be the important details, but I found the overall explanation incomplete and confusing. I believe that explaining where the activation patterns come from, what the criteria for "trainable" are, etc., would strengthen this diagram and its associated text.

---

> ### Author Response · Authors · 2026-06-25
>
> We sincerely appreciate the reviewer for the constructive suggestion. In the revised manuscript, we make Figure 4 and its accompanying discussion more explicit by clarifying the source and interpretation of the activation patterns. Specifically, we state that the activation patterns are representative forward activations collected from different candidate split positions, and that each panel compares the corresponding activations before and after 4-bit quantization.
>
> We also clarify what aspects of the visualization are relevant to trainability. For ReLU, the key observation is that the pre-quantization activations are naturally sparse, and this sparsity is largely preserved after 4-bit quantization. For SwiGLU and GELU, although the activations are denser, the quantized activations still retain sufficient informative nonzero elements to support effective training. By contrast, when the split is placed between transformer blocks, the 4-bit quantized activations become substantially distorted, which prevents effective fine-tuning in our experiments.
>
> In addition, we revise the caption and surrounding text to define the "trainable" label more precisely. A configuration is labeled trainable if, under the same fine-tuning setting, the model achieves non-trivial test accuracy above the random-choice baseline. It is labeled "untrainable" if the test accuracy remains near the random-choice baseline throughout training or close to zero.

---

### Review · Reviewer_NkKQ · 2026-06-18

**Summary Of Contributions:**

The paper proposes SparQ, a model-parallel framework combining zeroth-order optimization and quantization-induced activation sparsity for communication-efficient LLM fine-tuning. The method places split layers after activation functions and transmits sparse activations. Experiments demonstrate reductions in memory and communication costs. The main strengths are practical relevance and a simple framework design. The main weaknesses are limited novelty, weak large-scale evaluation, and insufficient theoretical justification.

**Audience:**

Yes

**Audience Explanation:**

The paper studies communication- and memory-efficient fine-tuning of large language models, which is an important topic for the machine learning systems and optimization communities. The empirical observations on quantization-induced sparsity may also be of independent interest.

**Broader Impact Concerns:**

I do not identify significant broader impact concerns beyond those already discussed by the authors. The work primarily focuses on improving the efficiency of distributed LLM fine-tuning.

**Claims And Evidence:**

No

**Claims Explanation:**

### Strengths
* **Relevant Problem Setting:** Addressing the memory and communication bottlenecks in LLM fine-tuning across distributed nodes is a highly practical and important research direction.
* **Interesting Empirical Observation:** The finding that 4-bit quantization induces pervasive sparsity even in naturally dense activations is a valuable insight.
* **Clear Presentation:** The paper is generally well-written, and Figure 3 effectively illustrates the core motivation behind the framework.

### Critical Weaknesses

**1. Theoretical Inconsistencies and Questionable Assumptions**
The theoretical contribution is fundamentally flawed due to a direct contradiction between the assumptions and the actual methodology. The authors claim an $\mathcal{O}(\sqrt{d/T})$ convergence rate. However, Assumption 2 explicitly requires the stochastic gradient to be unbiased: $\mathbb{E}[\hat{\nabla} f] = \nabla f$. In contrast, the paper defaults to the Forward Difference method for gradient estimation:

$$\hat{G} = \frac{f(x+\mu u)-f(x)}{\mu} \cdot u$$

This is a well-known *biased* estimator. Building a convergence theorem upon assumptions that are inconsistent with the actual algorithm invalidates the theoretical claims. Furthermore, Lemma 1 bounds the compression error, but the analysis fails to explicitly map how quantization error, sparsification error, and index encoding error propagate into the final optimization bounds.

**2. The PEFT Elephant in the Room (Outdated Baselines)**
The paper's entire motivation rests on alleviating the massive memory and communication costs of full-parameter fine-tuning. However, in the current landscape of LLM fine-tuning, Parameter-Efficient Fine-Tuning (PEFT) methods—such as LoRA, QLoRA, and GaLore—are the industry standards for exactly these bottlenecks. By omitting PEFT baselines entirely, the evaluation is misaligned with reality. If a standard FO-SGD method were equipped with LoRA in an MP setup, the memory and backward communication overhead would drop precipitously. Without a direct comparison to "FO + MP + LoRA" or recent ZO baselines (e.g., Sparse-MeZO), it is impossible to determine if SparQ offers any practical advantage in modern workflows.

**3. Misaligned System Evaluation and Trivial Scaling**
The paper repeatedly emphasizes "Model Parallelism," yet the core experiments are almost exclusively conducted on a trivial M=2 (two GPU) setup. Real-world MP deployments involve 8, 16, 32, or 64 GPUs across multiple nodes, where network topology and communication latency become critical. Relegating M>2 analysis to the appendix without empirical scaling curves is unacceptable for a systems-oriented paper.
Crucially, the communication cost is evaluated purely by *theoretical byte count*. ZO methods require multiple forward passes per step (e.g., P=5 perturbations). In distributed systems, the *frequency* of communication (latency) is often more detrimental than the *volume* (bandwidth). Substituting theoretical byte counts for actual wall-clock throughput (e.g., tokens/sec or samples/sec) masks the true system overhead of the proposed method.

**4. Performance Collapse and Insufficient Benchmarks**
The evaluation relies on outdated, small-scale datasets (SST-2, WIC, RTE) that do not reflect modern LLM capabilities. Even within these limited benchmarks, a dangerous signal emerges: while SparQ matches or slightly beats FO-SGD on SST-2 (a simple sentiment classification task), its performance collapses on tasks requiring deeper contextual understanding. For instance, on OPT-1.3B, FO-SGD achieves 63.5% on WIC and 70.8% on RTE, while SparQ plummets to 55.6% and 57.1%, respectively. This performance degradation indicates that the aggressive 4-bit quantization and forced sparsity irreversibly destroy complex representations in SwiGLU/GELU activations. Evaluating this method on modern reasoning benchmarks (e.g., MMLU, GSM8K, Instruction Tuning) is mandatory to prove the model remains functional.

**Requested Changes:**

## Critical Changes

The following issues are critical for acceptance:

1. **Evaluate on realistic model-parallel settings.**
   The current experiments are primarily conducted with two-way model partitioning. Additional evaluations on larger-scale settings (e.g., 8–32 GPUs) are necessary to validate the scalability claims of the proposed framework.

2. **Report end-to-end efficiency metrics.**
   The paper focuses on communication reduction but does not provide sufficient evidence regarding practical training efficiency. Wall-clock training time, throughput, and overall system speedup should be reported.

3. **Compare against stronger and more recent baselines.**
   The empirical evaluation should include recent zeroth-order and parameter-efficient fine-tuning methods, such as Sparse-MeZO and LoZO, to better establish the competitiveness of the proposed approach.

4. **Strengthen the theoretical justification.**
   The convergence analysis relies on assumptions whose relationship to the practical forward-difference estimator is not sufficiently clarified. The theoretical guarantees should be revised or better justified.

## Non-Critical Changes

The following suggestions would further strengthen the paper:

1. **Include more representative LLM benchmarks.**
   Evaluation on modern benchmarks such as MMLU and GSM8K would improve the assessment of practical applicability.

2. **Provide a more detailed ablation study.**
   Additional analysis of split-layer placement and its impact on communication, memory, and accuracy would offer deeper insight into the proposed design choices.

---

> ### Author Response · Authors · 2026-06-25
>
> We sincerely appreciate the reviewer for the constructive comments, and we would like to reply to that as follows.
>
> > Theoretical Inconsistencies and Questionable Assumptions | Strengthen the theoretical justification. .
>
> **Our response:** Thank you for the comment. The forward-difference estimator is indeed biased with respect to the original gradient $\nabla f(x)$. Yet, our convergence analysis follows the standard ZOoptimization framework based on the Gaussian-smoothed objective $f_\mu$, for which $E_u\left[\frac{f(x+\mu u)-f(x)}{\mu}u\right]=\nabla f_\mu(x)$. Thus, the proof does not require the forward-difference estimator to be an unbiased estimator of $\nabla f(x)$. The bias between $\nabla f_\mu(x)$ and $\nabla f(x)$ is explicitly bounded in Lemma 3 and controlled by the choice of $\mu$, which leads to the stated $O(\sqrt{d/T})$-type rate. Regarding compression, Assumption 5 models the activation compressor by $\|x-C(x)\|\le \kappa\|x\|$. Lemma 1 then maps this activation compression error into the gradient discrepancy:
> $\|\nabla f(x_t)-\nabla \hat f(x_t)\|^2
> \le (1+C_{S_1}^2)L_2^2\kappa^2L_{S_1}^2$. This term appears explicitly as the third term in Theorem 1. The index encoding itself is lossless, and the quantization/sparsification effect is captured by $\kappa$. We will clarify the notation around Assumption 2 to avoid the impression that the forward-difference estimator is assumed unbiased for $\nabla f$.
>
> > The PEFT Elephant in the Room.
>
> **Our response:** Thank you for the comment. PEFT methods such as LoRA/QLoRA/GaLore are important for efficient fine-tuning, but they target a different bottleneck from SparQ. LoRA/QLoRA mainly reduce the number of trainable parameters and optimizer states, while GaLore reduces optimizer memory through low-rank gradient projection. In contrast, SparQ targets the communication and activation-memory bottlenecks in model-parallel fine-tuning, especially cross-partition activation/gradient communication.
>
> Even with LoRA in a first-order MP setup, training still requires backpropagation through the model to update the adapters. Thus, intermediate activations must be stored, and backward activation gradients still need to be communicated across split boundaries. SparQ removes this high-dimensional backward communication by ZO optimization, where only scalar feedback is sent backward, and further reduces forward activation communication through quantization-induced sparse activation transmission. Recent ZO methods such as Sparse-MeZO and LoZO are also complementary rather than direct substitutes. Sparse-MeZO reduces the effective parameter dimension by perturbing/updating selected parameters, while LoZO improves ZO gradient estimation using low-rank structures. These methods could potentially serve as stronger base ZO optimizers inside SparQ, but they do not directly address the model-parallel activation communication bottleneck studied in this paper.
>
> Thus, FO+MP+LoRA is not a direct replacement for SparQ, and Sparse-MeZO/LoZO solve a different part of the efficiency problem. Our comparisons with FO-SGD, AQ-SGD, and MeZO/ZO-SGD are intended to isolate the effects of model-parallel communication, activation compression, and ZO optimization. PEFT and recent ZO variants are largely orthogonal to SparQ, and systematically combining them with SparQ is an interesting but separate direction beyond the scope of this optimization-focused study.

---

> ### Author Response · Authors · 2026-06-25
>
> > Misaligned System Evaluation and Trivial Scaling | Evaluate on realistic model-parallel settings..
>
> **Our response:** Thank you for the comment. Our work focuses on the optimization aspect of model-parallel zeroth-order fine-tuning, rather than cluster-level systems optimization. The main objective is to reduce memory usage and cross-partition communication volume while preserving the behavior of the underlying ZO fine-tuning method.
> We use $M=2$ since it is the minimal model-parallel setting that exposes the split-boundary communication bottleneck targeted by SparQ. The multi-partition case is also covered in the paper: Appendix B.2 gives the $M>2$ algorithm, and Appendix C.3 offers the corresponding convergence analysis. For the forward-difference variant, SparQ communicates $(P+1)(M-1)$ compressed activations and $P(M-1)$ scalar messages per step, while eliminating high-dimensional backward-gradient communication.
> Regarding latency, ZO methods require multiple forward evaluations, but first-order MP methods require backward propagation with high-dimensional gradient communication across split boundaries. Our communication-volume evaluation is intended to isolate the algorithmic communication cost targeted by SparQ. Wall-clock throughput on 8-64 GPUs depends heavily on hardware, interconnect, topology, and scheduling implementation. Due to our limited GPU resources, we do not have access to such a model-parallel GPU cluster, so reliable large-scale throughput measurements are not feasible. Thus, our evaluation focuses on the quantities directly optimized by the proposed method: memory usage and communication volume.
>
> > Report end-to-end efficiency metrics.
>
> **Our response:** Thank you for the comment. Our paper focuses on the optimization and communication aspects of model-parallel ZO fine-tuning, rather than full system-level throughput optimization. The primary quantities targeted by SparQ are memory usage and cross-partition communication volume, which are therefore the main efficiency metrics reported in the paper. End-to-end wall-clock time and throughput depend heavily on hardware, interconnect bandwidth/latency, network topology, software stack, and scheduling implementation. Since we do not have access to a large model-parallel GPU cluster, reliable system-level throughput measurements are not feasible for us. Reporting wall-clock numbers from a small local setup would not faithfully represent realistic large-scale MP efficiency. Instead, we report the hardware-independent quantities directly optimized by SparQ: memory usage and communication cost. SparQ reduces high-dimensional backward-gradient communication to scalar feedback and compresses forward activations through quantization-induced sparsity. These measurements isolate the algorithmic efficiency gains of SparQ, which is the focus of this optimization-oriented work.
>
> > Compare against stronger and more recent baselines.
>
> **Our response:** Thank you for the comment. Sparse-MeZO and LoZO are recent and relevant ZO fine-tuning methods, but they focus on a different aspect of the problem. Sparse-MeZO improves MeZO by applying ZO updates to a selected subset of parameters, while LoZO improves ZO gradient estimation through low-rank structures. Similarly, PEFT methods mainly reduce trainable parameters or optimizer memory. In contrast, SparQ targets the model-parallel setting, specifically reducing cross-partition activation communication and eliminating high-dimensional backward-gradient communication. Our comparison with MeZO/ZO-SGD is intended to isolate the effect of SparQ's split-layer placement and quantized sparse activation transmission. The results show that SparQ preserves the performance of the underlying ZO method while substantially reducing communication and memory costs. Sparse-MeZO, LoZO, and PEFT methods are largely orthogonal to SparQ: they could potentially replace or complement the base fine-tuning method inside SparQ, but they do not directly address the model-parallel activation communication bottleneck studied in this paper. Therefore, they are relevant complementary methods rather than direct substitutes for the main contribution of SparQ.

---

> ### Author Response · Authors · 2026-06-25
>
> > Provide a more detailed ablation study.
>
> **Our response:** Thank you for the suggestion. We have included ablations related to the main design choices of SparQ. Fig. 4 compares splitting immediately after activation functions with splitting between Transformer blocks, showing that split placement substantially affects the quality and compressibility of transmitted activations. This supports our choice of placing split layers at sparsity-rich activation outputs.
> The impact on communication is directly tied to the split location: after quantization, layers with lower nonzero activation ratios require fewer transmitted values and indices. This is why Fig. 3 reports layer-wise sparsity patterns and why we select split points near the middle block with high activation sparsity. The memory effect is mainly controlled by balancing the model partition across two GPUs; therefore, we choose split points near the middle while also considering sparsity.
> In addition, Table 1 studies different quantization levels and reports both accuracy and communication cost, while Table 2 reports accuracy, memory, and communication across models and datasets. Together, these results analyze the main tradeoff among split placement, quantization, communication, memory, and accuracy.